# Improving Fine-Grained Control via Aggregation of Multiple Diffusion Models

## Abstract

While many diffusion models perform well when controlling particular aspects such as style, character, and interaction, they struggle with fine-grained control due to dataset limitations and intricate model architecture design. This paper introduces a novel training-free algorithm for fine-grained generation, called Aggregation of Multiple Diffusion Models (AMDM). The algorithm integrates features in the latent data space from multiple diffusion models within the same ecosystem into a specified model, thereby activating particular features and enabling fine-grained control. Experimental results demonstrate that AMDM significantly improves fine-grained control without training, validating its effectiveness. Additionally, it reveals that diffusion models initially focus on features such as position, attributes, and style, with later stages improving generation quality and consistency. AMDM offers a new perspective for tackling the challenges of fine-grained conditional generation in diffusion models. Specifically, it allows us to fully utilize existing or develop new conditional diffusion models that control specific aspects, and then aggregate them using the AMDM algorithm. This eliminates the need for constructing complex datasets, designing intricate model architectures, and incurring high training costs.

## 1 Introduction

Diffusion models (Sohl-Dickstein et al., 2015; Ho et al., 2020; Song et al., 2021a;b; Karras et al., 2022) have achieved excellent performance in generative tasks. In particular, conditional diffusion models (Rombach et al., 2022; Chung et al., 2023; Esser et al., 2024) not only deliver advanced results in practical applications such as Text-to-Image (T2I) (Nichol et al., 2022; Chen et al., 2023; Lee et al., 2024; Xu et al., 2024) and Image-to-Image (I2I) generation (Zhang et al., 2023; Mou et al., 2024), but also offer highly flexible conditional control mechanisms.

Recent research on conditional diffusion models has focused on achieving fine-grained control, including object attributes (Wu et al., 2023; Wang et al., 2024a), interactions (Hoe et al., 2024; Jia et al., 2024), layouts (Zheng et al., 2023; Chai et al., 2023; Chen et al., 2024b), and style (Wang et al., 2023; Huang et al., 2024; Qi et al., 2024). However, maintaining consistency across diverse nuanced control remains a significant challenge. Generating multiple objects with overlapping bounding boxes can lead to attribute leakage, where the description of one object inappropriately influences others, causing inconsistencies between objects and the background. Fine-grained interaction details may be illogical, and style integration may compromise object attributes.

Existing generation approaches only partially address these issues due to the inherent complexity and diversity of fine-grained control, coupled with limitations in datasets and model architectures. Some works (Li et al., 2023; Zhou et al., 2024; Wang et al., 2024b) may perform well in preventing attribute leakage among multiple instances during layout generation but perform poorly in managing object interactions, while others (Ye et al., 2023; Huang et al., 2024) may excel in style transfer but exhibit limited control over layout.

Interestingly, most of these methods are based on Stable Diffusion (Rombach et al., 2022), which is theoretically grounded in DDPM (Ho et al., 2020) and classifier-free guidance (Dhariwal & Nichol, 2021) for conditional control. In this context, a distinct line of research has recently emerged that leverages the shared diffusion equation across different models to combine multiple diffusion models

at inference time, thereby integrating their respective information. The predominant strategy is to apply linear weighting in the score space (Garipov et al., 2023; Bradley et al., 2025; Thornton et al., 2025; Skreta et al., 2025a; He et al., 2025; Skreta et al., 2025b). However, these methods were originally designed to synthesize new data distributions under mutually independent (orthogonal) conditioning variables (Bradley et al., 2025). Although fine-grained generation can be pursued by enforcing identical conditions, this practice violates the orthogonality assumption embedded in the original design, leading to suboptimal generation quality. To mitigate the distributional shift caused by linear weighting, several works have proposed more sophisticated weighting schemes, which, however, introduce additional inference overhead (Skreta et al., 2025b; He et al., 2025). Another line of methods performs linear weighting in the parameter space of the networks (Biggs et al., 2024; Oh et al., 2025; Wang et al., 2025), but this requires the combined models to share exactly the same architecture, thereby substantially limiting their applicability.

This paper proposes a training-free AMDM algorithm that operates directly in the latent data space guided by a geometric perspective. The algorithm is both conceptually simple and practically effective, relying solely on closed-form computations with negligible computational overhead. As a result, it offers an efficient and viable solution for fine-grained control tasks that are highly coupled and non-orthogonal in nature. In summary, this work makes the following contributions: **(1)** We propose a novel diffusion model aggregation algorithm, AMDM, that can aggregate intermediate variables in latent data space from multiple conditional diffusion models within the same ecosystem enabling fine-grained generation; **(2)** We conduct extensive experiments, with both qualitative and quantitative results demonstrating significant improvements, particularly in regions where previous models exhibited limited controllability; **(3)** We reveal that diffusion models initially focus on generating coarse-grained features such as position, attributes, and style, while later stages emphasize quality and consistency.

## 2 PRELIMINARIES: STABLE DIFFUSION

Stable Diffusion (Rombach et al., 2022) is a class of diffusion models defined in the latent space. The original data $\mathbf{x}_0$ is mapped through the encoder (VAE (Kingma & Welling, 2013)) to obtain the latent variable $\mathbf{z}_0 = E(\mathbf{x}_0)$, which evolves according to the DDPM (Ho et al., 2020) diffusion paradigm:
$$p(\mathbf{z}_t|\mathbf{z}_{t-1}) = \mathcal{N}(\sqrt{\alpha_t}\mathbf{z}_{t-1}, (1-\alpha_t)\boldsymbol{I}), \tag{1}$$
where $\mathbf{z}_t$ represents the noisy latent data at timestep $t \in [0, T]$, $\alpha_t$ is the coefficient drift schedule generally satisfying $\lim_{t \to T} \alpha_t = 0$. From (1), the forward marginal distribution is:
$$p(\mathbf{z}_t|\mathbf{z}_0) = \mathcal{N}(\sqrt{\bar{\alpha}_t}\mathbf{z}_0, (1-\bar{\alpha}_t)\boldsymbol{I}), \tag{2}$$
where $\bar{\alpha}_t = \prod_{i=1}^{t} \alpha_i$. Assuming the denoising neural network is $\boldsymbol{\epsilon}_\theta$ and the condition is $y$, the loss function is the variational lower bound of its likelihood, i.e., the KL divergence of the joint probability:
$$\mathcal{L} = KL(p(\mathbf{z}_{0:T})\|p_\theta(\mathbf{z}_{0:T})) \propto \mathbb{E}_{t,\mathbf{z}_t,\boldsymbol{\epsilon}_t}\left[\|\boldsymbol{\epsilon}_t - \boldsymbol{\epsilon}_\theta(\mathbf{z}_t, t, y)\|^2\right], \tag{3}$$
where $\boldsymbol{\epsilon}_t \sim \mathcal{N}(\mathbf{0}, \boldsymbol{I})$. More generally, the reverse sampling process is as follows:
$$p_\theta(\mathbf{z}_{t-1} \mid \mathbf{z}_t, y) = \mathcal{N}(\boldsymbol{\mu}_\theta(\mathbf{z}_t, t, y), \sigma_t^2 \boldsymbol{I})$$
$$= \mathcal{N}\left(\sqrt{\frac{\bar{\alpha}_{t-1}}{\bar{\alpha}_t}}\mathbf{z}_t + \left(\sqrt{1 - \bar{\alpha}_{t-1} - \sigma_t^2} - \sqrt{\frac{\bar{\alpha}_{t-1}(1-\bar{\alpha}_t)}{\bar{\alpha}_t}}\right)\boldsymbol{\epsilon}_\theta(\mathbf{z}_t, t, y), \sigma_t^2 \boldsymbol{I}\right), \tag{4}$$
where $\sigma_t$ is a free variable.

## 3 METHOD

In this section, we first analyze the challenges and limitations faced by current research on fine-grained control. We then discuss the existence and applicability of aggregation operations. Subsequently, we present Proposition 3.1, which reveals the geometric properties of the inference process, and Proposition 3.2, which guides the final design of AMDM.

### 3.1 ANALYSIS

Current fine-grained conditional control models offer limited controllability and face numerous issues. For example, given the caption "A red hair girl is drinking from a blue bottle of water, oil painting" and corresponding bounding boxes for positioning control, different models are likely to show varying performance, as illustrated in Figure 1. Model A, which receives additional inputs for position information and actions, excels at accurate positioning and interactions, achieving high-quality results. However, it struggles with attribute control and maintaining the oil painting style. Conversely, Model B incorporates extra input for position and attribute information, managing both but not accurately capturing interactions and stylistic elements. Model C references the style of an image, enabling precise management of style characteristics but lacking adequate control over location and attribute details. The fundamental reason for these issues lies in the complexity and flexibility of fine-grained control tasks, which makes it challenging for limited datasets and specific model architectures to account for all the intricate features. It is noteworthy that these models share a common foundation, as they are all based on the same diffusion process. Recognizing this shared basis, our objective is to develop an aggregation algorithm that leverages these commonalities to integrate the distinctive characteristics of multiple models, achieving fine-grained conditional control in a more direct and efficient manner.

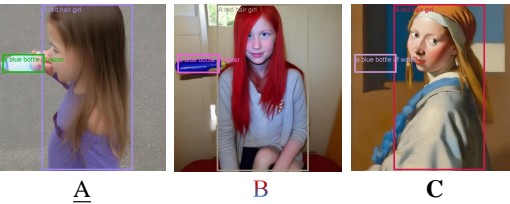

*Caption:* A **red** hair girl is drinking a **blue** bottle of water, **oil painting**

A      B      **C**

Figure 1: Examples of fine-grained conditional control of the same caption by different models.

**Do aggregation operations exist in different diffusion models?** For a latent diffusion model $p_\theta$, the true latent data manifold along the diffusion process is $\mathcal{M}_t(\xi_t) = \{\mathbf{z}_t \in \mathbb{R}^n \mid p(\mathbf{z}_t) \geq \xi_t\}$, where an appropriate choice of $\xi_t$ ensures that $\mathcal{M}_t$ is a set of meaningful data. We define the generation domain at time $t$ under condition $y$ as $D_{t,y}^\theta(\tau_{t,y}^\theta) = \{\mathbf{z}_t \in \mathbb{R}^n \mid p_\theta(\mathbf{z}_t \mid y) \geq \tau_{t,y}^\theta\}$. Since conditional samples are a selection of the unconditional population, there exists a largest threshold $\tau_{t,y}^\theta$ such that $D_{t,y}^\theta(\tau_{t,y}^\theta) \subset \mathcal{M}_t(\xi_t)$. This implies that, given $\xi_t$, the $D_{t,y}^\theta(\tau_{t,y}^\theta)$ is uniquely determined, and we thus omit the explicit dependence on the threshold for simplicity. Mathematically, the most fundamental requirement for performing aggregation operations is that all elements must reside in a shared space. However, in the case of aggregation between two models, it is evident that data obtained through conditional sampling from $p_{\theta_1}$ should lie within $D_{0,y_1}^{\theta_1}$, whereas data obtained from $p_{\theta_2}$ should lie within $D_{0,y_2}^{\theta_2}$. According to the definition, $D_{0,y_1}^{\theta_1} \subset \mathcal{M}_0, D_{0,y_2}^{\theta_2} \subset \mathcal{M}_0$, which implies that the data from different models can be embedded into a shared representation space $\mathcal{M}_0$. This observation directly leads to the **first requirement**: $\mathcal{M}_0$ must coincide across models, implying that the latent space encoders are consistent, and the aggregation operations at $t = 0$ exist. For $t > 0$, this naturally leads to the **second requirement**: $\mathcal{M}_t$ must coincide across models. In the context of diffusion models, a sufficient condition for this requirement is that the underlying SDEs are identical. Consequently, we obtain $D_{t,y_1}^{\theta_1}, D_{t,y_2}^{\theta_2} \subset \mathcal{M}_t$, which ensures the existence of aggregation operations for all time steps $t$.

**Which models can achieve fine-grained generation through aggregation operations?** In practice, we expect the result of the operation to lie within the smaller space $D_{t,y_1}^{\theta_1} \cap D_{t,y_2}^{\theta_2}$, since only in this space is it possible to faithfully incorporate information from both models and achieve fine-grained generation. In order to achieve aggregation operation at time $t$, it is further required that the domains $D_{t,y_1}^{\theta_1}$ and $D_{t,y_2}^{\theta_2}$ of different models possess a non-empty intersection. First, we observe that application models developed on top of SD, in addition to satisfying the aforementioned two theoretical requirements, share several common characteristics in their denoising networks: 1. Additive Architectures: Typically freeze most or all of the weights of the original base model and then integrate a new, lightweight, and trainable network module in parallel. 2. Modified Architectures: Replace or alter certain layers within the U-Net while retaining its overall structure. 3. Preserved Architectures: Fine-tuning is performed with additional training samples from specific domains while keeping the overall architecture intact. This indicates that although the three approaches differ in their denoising network architectures, they preserve most of the original functionality and only enhance specific control aspects. Therefore, we consider these models with the above properties to belong to the same **diffusion model ecosystem**, indicating that when given the same input, different

models exhibit similar tendencies. Accordingly, we can believe that there exists *functional proximity*: $\|\boldsymbol{\epsilon}_{\theta_1}(\mathbf{z}_t, t, y) - \boldsymbol{\epsilon}_{\theta_2}(\mathbf{z}_t, t, y)\| \leq L_t$. When it comes to fine-grained generation, the conditions $y_1$ and $y_2$ describe the same task $\mathcal{T}$, which means that they are strongly coupled and semantically consistent, as their global prompts are at least aligned. This implies that we can assume *conditional proximity*: $|y_1 - y_2| \leq L_{\mathcal{T}}$. Based on this, it is natural to infer that for two models $p_{\theta_1}$ and $p_{\theta_2}$ within the same diffusion model ecosystem, given two distinct condition descriptions $y_1$ and $y_2$ of a fine-grained task, they are inclined to generate the same samples and pursue a unified objective. Specifically, we have $D_{0,y_1}^{\theta_1} \cap D_{0,y_2}^{\theta_2} \neq \varnothing$. Moreover, for the noised state at any time $t$, we likewise have $D_{t,y_1}^{\theta_1} \cap D_{t,y_2}^{\theta_2} \neq \varnothing$. Therefore, models within the same diffusion ecosystem are capable of achieving fine-grained generation.

The above analysis suggests that an aggregation operation exists only when the models are within the same diffusion model ecosystem. This finding also provides theoretical support for the validity of other compositional methods and thus establishes a solid foundation for the design of the subsequent algorithm.

## 3.2 Algorithm: AMDM

**Spherical Aggregation.** To identify a feasible aggregation algorithm, we need to examine the properties of $\mathcal{M}_t$. At the initial stage $t = T$, since $p(\mathbf{z}_T)$ follows a standard Gaussian prior, it follows from the spherical concentration property of high-dimensional Gaussian distributions that $\mathcal{M}_T$ is concentrated on an $(n-1)$-dimensional manifold, namely an $n$-dimensional hypersphere. The proof is provided in Appendix A. For $t < T$, Chung et al. (2022) demonstrated that $\mathcal{M}_t$ is concentrated on an $(n-1)$-dimensional manifold, which approximates an $n$-dimensional hypersphere as $t$ becomes large. Furthermore, we have the following proposition:

**Proposition 3.1.** *Assume that the aggregated variable at time step $t$ is $\mathbf{z}_t'$. For the sampling step from $t$ to $t-1$, two diffusion models $p_{\theta_1}$ and $p_{\theta_2}$ sample $\mathbf{z}_{t-1}^{\theta_1}$ and $\mathbf{z}_{t-1}^{\theta_2}$ respectively from (4). Then the approximate upper bound for the absolute value of the difference of the norms is:*

$$\left| \|\mathbf{z}_{t-1}^{\theta_1}\| - \|\mathbf{z}_{t-1}^{\theta_2}\| \right| \leq \sqrt{1 - 2\alpha(1 - \cos\varphi)}\delta, \quad \cos\varphi \geq 1 - \frac{\delta^2}{\|\mathbf{z}_{t-1}^{\theta_1}\|^2 + \|\mathbf{z}_{t-1}^{\theta_2}\|^2}, \quad (5)$$

*where $\varphi$ is the angle between $\mathbf{z}_{t-1}^{\theta_1}$ and $\mathbf{z}_{t-1}^{\theta_2}$, $\alpha \in [0, 1/4]$, $\delta \leq B_t(L_{\theta_1} L_{\mathcal{T}} + L_t) + \sigma_t\sqrt{2n}$, $L_{\theta_1}$ is the Lipschitz constant, and $B_t$ is the coefficient of $\epsilon$ in the sampling process (4).*

The proof is provided in Appendix B and the numerical experiments are presented in Appendix D.2. From Proposition 3.1, it follows that the key factor governing the bound is $\delta$, where the first term arises from the conditional proximity and functional proximity in fine-grained tasks within the same diffusion model ecosystem, typically taking a relatively small value. For the second term, if the sampling noise is sufficiently small, particularly in the case of DDIM sampling, its effect can be neglected. Therefore, Proposition 3.1 indicates that when different models from the same diffusion ecosystem perform DDIM sampling on the same point in fine-grained tasks, the resulting new variables have approximately equal norms and small angles between them. Taken together, we may assert that when $t < T$, the sampled variables lie on a local spherical manifold. Motivated by the geometric properties of the global sphere at time $T$ and the local spheres at $t < T$, we propose spherical interpolation for aggregation, which maximizes the retention of aggregated data on the original manifold while minimizing deviations:

$$\begin{aligned} \mathbf{z}_{t-1}' &= Slerp(\mathbf{z}_{t-1}^{\theta_1}, \mathbf{z}_{t-1}^{\theta_2}, w) \\ &= \frac{\sin((1-w)\varphi)}{\sin(\varphi)}\mathbf{z}_{t-1}^{\theta_1} + \frac{\sin(w\varphi)}{\sin(\varphi)}\mathbf{z}_{t-1}^{\theta_2}, \end{aligned} \quad (6)$$

where $w \in [0, 1]$ is the weighting factor that balances the contribution of each model. Spherical aggregation integrates the conditional control information of $p_{\theta_2}$ and $p_{\theta_1}$, while keeping the new variables stable near the manifold.

**Deviation Optimization.** Ideally, we expect $\mathbf{z}_{t-1}' \in D_{t-1,y_1}^{\theta_1} \cap D_{t-1,y_2}^{\theta_2}$, but deviations are likely to occur in practice. Considering the step from $t$ to $t-1$, since $p_{\theta_1}(\mathbf{z}_{t-1}^{\theta_1} \mid \mathbf{z}_t^{\theta_1}, y_1)$ is high-dimensional Gaussian, its samples concentrate in a thin spherical shell around the mean. Motivated

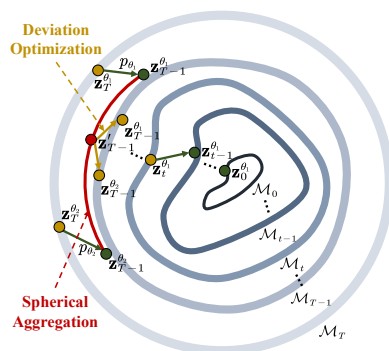 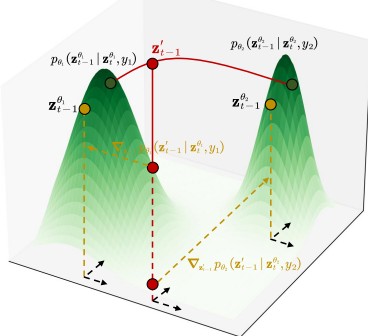

Figure 2: Geometry of AMDM (Left) and Deviation Optimization (Right). The algorithm employs spherical aggregation and deviation optimization to incorporate conditional information during the initial steps. Subsequently, direct sampling is applied to expedite the process and generate high-quality images.

by this, we perform a radial adjustment of $\mathbf{z}'_{t-1}$ toward that shell to obtain $\tilde{\mathbf{z}}_{t-1} \in D^{\theta_1}_{t-1,y_1}$. This adjustment alone does not guarantee $\tilde{\mathbf{z}}_{t-1} \in D^{\theta_2}_{t-1,y_2}$. Nevertheless, we show that $\tilde{\mathbf{z}}_{t-1}$ belongs to $D^{\theta_2}_{t-1,y_2}$ with high probability, which leads to the following proposition:

**Proposition 3.2.** *For the diffusion model $p_{\theta_1}$ defined by (4) and any new intermediate variable $\mathbf{z}'_{t-1}$ from (6), let:*

$$\tilde{\mathbf{z}}_{t-1} = \mathbf{z}'_{t-1} - \eta^{\theta_1}_{t-1} \frac{\mathbf{z}'_{t-1} - \boldsymbol{\mu}_{\theta_1}(\mathbf{z}^{\theta_1}_t, t, y_1)}{\|\mathbf{z}'_{t-1} - \boldsymbol{\mu}_{\theta_1}(\mathbf{z}^{\theta_1}_t, t, y_1)\|}, \tag{7}$$

*where $\eta^{\theta_1}_{t-1}$ is a small optimization step size. There exists $\eta^{\theta_1}_{t-1}$ such that $\tilde{\mathbf{z}}_{t-1} \in D^{\theta_1}_{t-1,y_1}$. Moreover, an approximate lower bound on the probability that $\tilde{\mathbf{z}}_{t-1} \in D^{\theta_2}_{t-1,y_2}$ is given by:*

$$P\left(\tilde{\mathbf{z}}_{t-1} \in D^{\theta_2}_{t-1,y_2}\right) \geq 1 - 2\exp\left(-\frac{n\left(\epsilon^{\theta_2}_{t-1} - \frac{d}{\sigma_t\sqrt{n}}\right)^2}{1 + 2\left(\epsilon^{\theta_2}_{t-1} - \frac{d}{\sigma_t\sqrt{n}}\right)}\right), \tag{8}$$

*where $d = \phi_w(\varphi)\|\mathbf{z}^{\theta_1}_{t-1} - \mathbf{z}^{\theta_2}_{t-1}\| + \eta^{\theta_1}_{t-1}$ and $\phi_w(\varphi) = \sin\big((1-w)\varphi/2\big)/\sin(\varphi/2)$.*

The proof is given in Appendix C; a detailed analysis of $d$ is presented in Appendix D.3; and the geometry of deviation optimization is illustrated in Figure 2 (Right). Proposition 3.2 shows that $\tilde{\mathbf{z}}_{t-1}$ obtained after spherical aggregation can be corrected into $D^{\theta_1}_{t-1,y_1}$ through deviation optimization, and with high probability can also be corrected into $D^{\theta_2}_{t-1,y_2}$. This not only improves the quality of sampling but also establishes that the new variable is able to simultaneously capture information from both models.

Combining equations (6) and (7), the algorithm for aggregation of two diffusion models is presented, comprising two key components: spherical aggregation and deviation optimization. Spherical aggregation aggregates the conditional control information from different models and ensures that the new intermediate variables remain stable near the manifold, while deviation optimization ensures more precise retention on the corresponding data manifold, enhancing sample quality. The algorithm iteratively performs spherical aggregation and deviation optimization for each model during the first $s$ steps, followed by direct sampling from $p_{\theta_1}$. This algorithm can be readily extended to multiple models, resulting in the final Aggregation of Multiple Diffusion Models (AMDM) algorithm, as shown in Algorithm 1 and Figure 2 (Left), where the spherical aggregation of multiple elements is defined through successive pairwise aggregation.

We perform aggregation only in the first $s$ steps, and subsequently apply a single model $p_{\theta_1}$ for inference. This procedure can be regarded as aggregating the features of other models into $p_{\theta_1}$, striking a balance between efficiency and effectiveness. Moreover, note that since $\mu_{\theta_i}(\mathbf{z}^{\theta_i}_t, t, y_i)$ can reuse $\epsilon_{\theta_i}(\mathbf{z}^{\theta_i}_t, t, y_i)$ from the previous sampling step, the deviation optimization introduces only a

---

**Algorithm 1** AMDM

**Input:** models from the same diffusion ecosystem $p_{\theta_1}, p_{\theta_2}, ..., p_{\theta_N}$, conditions $y_1, y_2, ..., y_N$, aggregation step $s$, weighting factors $w_1, w_2, ..., w_{N-1}$ and optimization steps $\eta_t^{\theta_1}, \eta_t^{\theta_2}, ..., \eta_t^{\theta_N}$

$\mathbf{z}_T^{\theta_1}, \mathbf{z}_T^{\theta_2}, ..., \mathbf{z}_T^{\theta_N} \sim N(\mathbf{0}, \boldsymbol{I})$

**for** $t = T, ..., 1$ **do**

    $\mathbf{z}_{t-1}^{\theta_1} \sim p_{\theta_1}(\mathbf{z}_{t-1}^{\theta_1} | \mathbf{z}_t^{\theta_1}, y_1)$

    **if** $t > T - s$ **then**

        $\mathbf{z}_{t-1}^{\theta_i} \sim p_{\theta_i}(\mathbf{z}_{t-1}^{\theta_i} | \mathbf{z}_t^{\theta_i}, y_i), i \in [2, N]$

        $\mathbf{z}_{t-1}' = Slerp(\mathbf{z}_{t-1}^{\theta_1}, ..., \mathbf{z}_{t-1}^{\theta_N}, w_1, ..., w_{N-1})$

        $\mathbf{z}_{t-1}^{\theta_i} = \mathbf{z}_{t-1}' - \eta_{t-1}^{\theta_i} \frac{\mathbf{z}_{t-1}' - \mu_{\theta_i}(\mathbf{z}_t^{\theta_i}, t, y_i)}{\|\mathbf{z}_{t-1}' - \mu_{\theta_i}(\mathbf{z}_t^{\theta_i}, t, y_i)\|}, i \in [1, N]$

    **end if**

**end for**

$\mathbf{x}_0^{\theta_1} = \text{Decoder}(\mathbf{z}_0^{\theta_1})$

**Output:** $\mathbf{x}_0^{\theta_1}$

---

single mathematical operation with negligible computational overhead, further reducing inference time.

As for the selection of $p_{\theta_1}$, in order to ensure high-quality generation in the later stages, it is natural to choose the model with stronger generative capability as $p_{\theta_1}$. This choice can be evaluated using a variety of criteria, such as quantitative metrics, model size, application universality, or direct experimental validation.

## 4 EXPERIMENTS

In this section, we aggregate several classic conditional diffusion models, all of which belong to the same Stable Diffusion ecosystem, followed by a series of ablation experiments to demonstrate its optimality. The core idea of our experiment is to demonstrate the effectiveness of the AMDM algorithm: Given a set of models with varying control capabilities, we only need to focus on whether the target model, after aggregating features from other models, can significantly enhance its performance in areas of weak control, while simultaneously introducing only minor performance trade-offs in its areas of strength. All experiments were conducted using a single RTX 3090 GPU and experimental details and additional experiments are provided in Appendix F.

### 4.1 AGGREGATION EXPERIMENTS

**InteractDiffusion and MIGC.** InteractDiffusion (Hoe et al., 2024) is a T2I model that combines a pretrained Stable Diffusion (SD) model with a locally controlled interaction mechanism, enabling fine-grained control over the generated images and demonstrating effective interactivity. MIGC (Zhou et al., 2024) is a T2I model that employs a divide-and-conquer strategy, achieving excellent performance in

Table 1: Quantitative results on the HOI Detection Score, FID and CLIP Score across different models.

| Method | Default ↑ | | Known Object ↑ | | FID ↓ | CLIP Score ↑ |
|---|---|---|---|---|---|---|
| | Full | Rare | Full | Rare | | |
| MIGC | 16.87 | 18.05 | 17.84 | 19.02 | 30.53 | 27.36 |
| MIGC(+InteractDiffusion) | **26.04** | **21.73** | **27.02** | **22.89** | **22.32** | **27.55** |
| InteractDiffusion | 29.53 | 23.02 | 30.99 | 24.93 | 18.69 | 26.91 |
| InteractDiffusion(+MIGC) | 31.40 | 24.52 | 32.76 | 26.32 | 18.35 | 27.18 |

both attribute representation and isolation of generated instances. InteractDiffusion primarily focuses on controlling subject-object interactions. However, due to the lack of explicit constraints on object attributes within the model architecture and dataset design, it exhibits suboptimal performance in attribute control. To address this, we attempt to aggregate the features of MIGC $p_{\theta_2}$ into InteractDiffusion $p_{\theta_1}$ by applying the AMDM algorithm, introducing attribute control information, and denoting this as InteractDiffusion(+MIGC).

Visual results are shown in Figure 3a. It is evident that aggregating the MIGC model into InteractDiffusion using our proposed AMDM algorithm significantly enhances its learned representations,

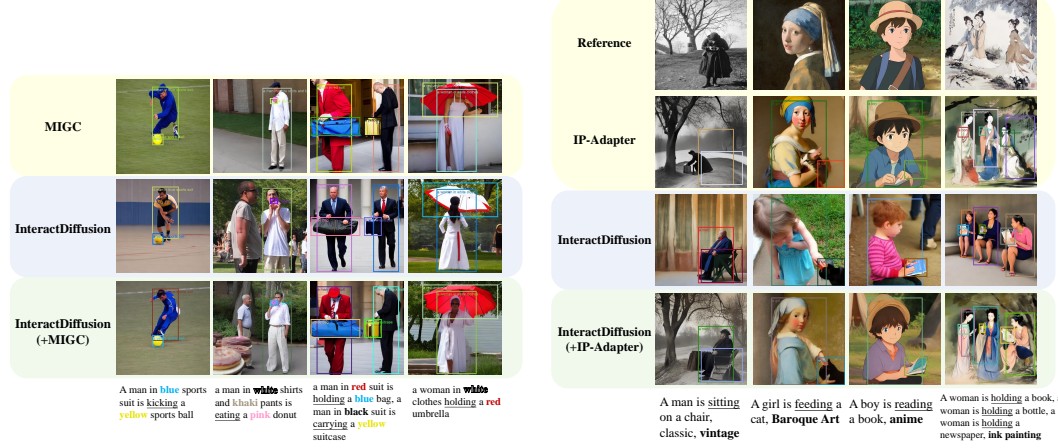

(a) MIGC into InteractDiffusion.  (b) IP-Adapter into InteractDiffusion.

Figure 3: Visual results of applying AMDM algorithm.

Table 2: Quantitative results on the COCO-MIG benchmark and CLIP Score across different models.

| Method | Instance Success Rate (%) ↑ | | | | | | mIoU Score (%) ↑ | | | | | | CLIP Score ↑ | |
| --- | --- | --- | --- | --- | --- | --- | --- | --- | --- | --- | --- | --- | --- | --- |
| | $L_2$ | $L_3$ | $L_4$ | $L_5$ | $L_6$ | Avg | $L_2$ | $L_3$ | $L_4$ | $L_5$ | $L_6$ | Avg | Global | Local |
| InteractDiffusion | 37.50 | 35.62 | 35.31 | 30.62 | 34.16 | 34.06 | 32.98 | 31.63 | 30.82 | 28.29 | 30.40 | 30.40 | 31.09 | 27.56 |
| InteractDiffusion(+MIGC) | 62.29 | 54.33 | 56.31 | 53.87 | 52.89 | 54.78 | 54.30 | 46.37 | 48.44 | 47.65 | 46.64 | 47.74 | 32.81 | 28.96 |
| MIGC | 64.06 | 56.04 | 58.43 | 56.00 | 49.89 | 55.46 | 54.43 | 49.33 | 50.48 | 48.67 | 44.74 | 48.53 | 32.78 | 28.61 |
| MIGC(+InteractDiffusion) | 60.00 | 50.20 | 50.46 | 48.25 | 46.97 | 49.78 | 52.58 | 44.27 | 43.39 | 42.67 | 41.50 | 43.69 | 32.41 | 28.55 |

leading to a notable improvement in instance attribute control, and confirming the algorithm's effectiveness.

MIGC demonstrates strong attribute control on the COCO-MIG benchmark (Zhou et al., 2024), while InteractDiffusion mainly uses the FGAHOI (Ma et al., 2023) for Human-Object Interaction (HOI) detection to show its control over interaction. To further validate the effectiveness of AMDM, we evaluate whether MIGC(+InteractDiffusion) can significantly enhance HOI interaction, ideally reaching the performance level of InteractDiffusion while introducing only minor trade-offs in attribute control on COCO-MIG. Conversely, we also examine whether InteractDiffusion(+MIGC) meets the experimental objectives.

The quantitative results of InteractDiffusion(+MIGC) are shown in the first two rows of Table 2. It can be observed that all metrics have significantly improved in InteractDiffusion(+MIGC) in attribute control. Interestingly, as shown in the last two rows of Table 1, it even surpasses the original InteractDiffusion model in terms of interaction. Similarly, the first two rows of Table 1 and the last two rows of Table 2 show that MIGC(+InteractDiffusion) achieves significant improvements across all metrics in interaction capability control, while incurring only minor trade-offs in attribute control, thereby validating the effectiveness of the algorithm.

**InteractDiffusion and IP-Adapter.**  IP-Adapter (Ye et al., 2023) is a lightweight I2I model that employs a decoupled cross-attention mechanism to separately process text and image features, enabling multimodal image generation. Due to its superior performance in preserving the style of the reference image, we propose integrating the style information from IP-Adapter $p_{\theta_3}$ into InteractDiffusion $p_{\theta_1}$, denoted as InteractDiffusion(+IP-Adapter). The experimental results are shown in Figure 3b. It can be observed that IP-Adapter enhances the style representations of InteractDiffusion, fully activating its style controllability, which further validates the effectiveness of the algorithm.

**InteractDiffusion, MIGC and IP-Adapter.**  Furthermore, we attempt to aggregate the attribute features from MIGC $p_{\theta_2}$ and the style features from IP-Adapter $p_{\theta_3}$ into InteractDiffusion $p_{\theta_1}$ to evaluate the effectiveness of the AMDM algorithm. The experimental results are presented in Figure 4.

## 4.2 ABLATION STUDIES

We first attempt a comparison with linear aggregation of InteractDiffusion(+MIGC) which has been of broad interest in compositional generation and the results are shown in Figure 5a. It can be observed that when the number of aggregation steps is small ($s < 5$), linear aggregation performs slightly better than spherical aggregation. However, as the number of aggregation steps increases, spherical aggregation significantly outperforms linear aggregation across various metrics. This is because, with more aggregation steps, the cumulative error due to the deviation in the data manifold becomes increasingly larger in linear aggregation, which negatively impacts the image quality. As shown in Figure 5b, the characteristic of manifold deviation in linear aggregation is evident. In contrast, spherical aggregation consistently minimizes the deviation of the aggregated variables from the spherical manifold, preserving the quality of the final image as much as possible. The deviation optimization further enhances the image quality, demonstrating its effectiveness. A more detailed discussion of linear compositional methods can be found in the Related Work and Appendix E.

We conduct experimental analyses on the absolute value of the norm difference, the angle, and $\delta$, thereby validating the correctness of Proposition 3.1. The results are provided in Appendix D.2 and Appendix D.3. Moreover, as shown in Figure 5a, the effectiveness of spherical aggregation consistently increases over time and remains almost entirely on the manifold, further validating our inference regarding the local spherical manifold in spherical aggregation.

We also investigated the optimization step of deviation optimization, with experimental results shown in Table 4. Obviously, when the optimization step is set to 0.3, the best performance is achieved across all metrics. Moreover, the results outperform those achieved with 50 aggregation steps without optimization.

Finally, we conduct ablation studies on the aggregation at different stages of the AMDM algorithm. As shown in Table 3, the best results are achieved when aggregation occurs during the initial stages of sampling. At other stages, especially the final ones, this leads to a noticeable drop in generation quality, with weaker control over attributes and interactions. This further supports the claim that diffusion models initially focus on generating features such as position, attributes, and style, while later stages emphasize quality and consistency.

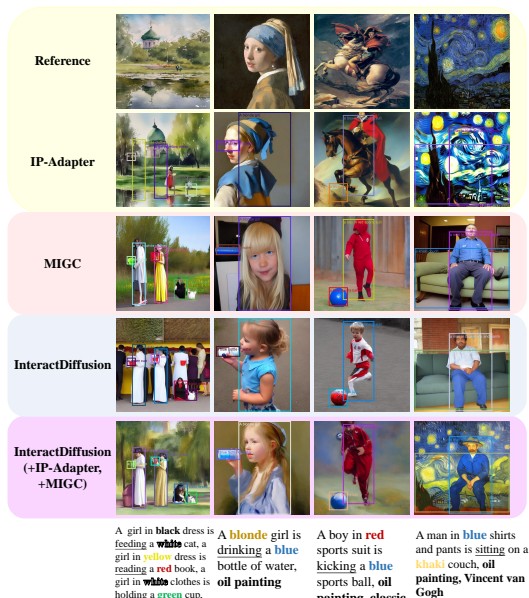

Figure 4: Visual results of aggregating MIGC and IP-Adapter into InteractDiffusion.

Table 3: InteractDiffusion(+MIGC) with total 20 steps sampling and $s=10$.

| Aggregation Stage | ISR (avg) ↑ | Default (Full) ↑ | FID ↓ |
|---|---|---|---|
| $t_{20} \to t_{10}$ | **54.78** | **31.40** | **18.35** |
| $t_{15} \to t_5$ | 46.23 | 26.21 | 27.45 |
| $t_{10} \to t_0$ | 41.65 | 23.47 | 44.62 |

## 5 RELATED WORK

**Fine-Grained Generation.** Beyond text-driven models (Nichol et al., 2022; Ramesh et al., 2022; Li et al., 2024b; Podell et al., 2024), research is increasingly moving toward finer-grained conditional control. A classical route is personalization-controlled generation—covering style (Sohn et al., 2023; Hertz et al., 2024; Chen et al., 2024a), subject (Li et al., 2024a; Shi et al., 2024; Jiang et al., 2024; Ye et al., 2023), person (Xiao et al., 2024; Giambi & Lisanti, 2023; Peng et al., 2024), and interactive generation (Huang et al., 2023b; Guo et al., 2024; Hoe et al., 2024). Another focus is spatial control (Li et al., 2023; Cheng et al., 2023; Nie et al., 2024; Zhou et al., 2024), which uses

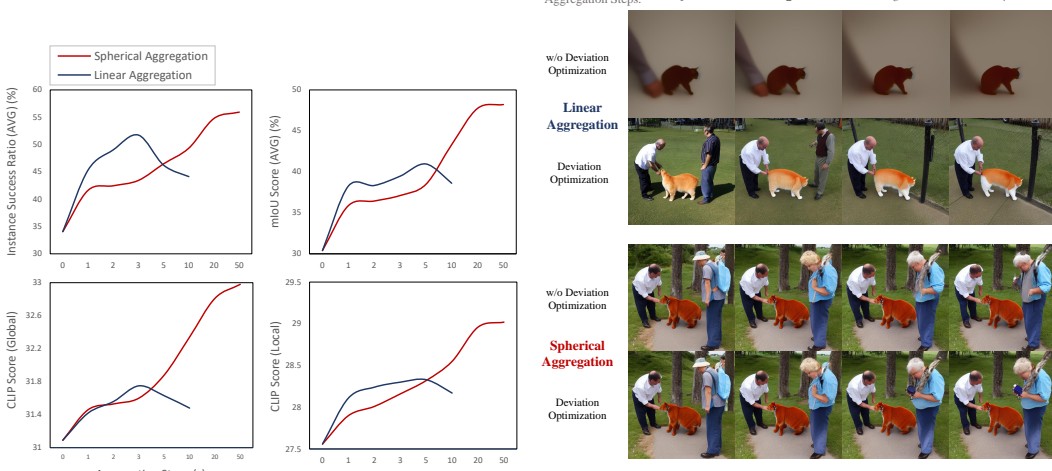

(a) Metrics of spherical aggregation and linear aggregation under different aggregation steps.

(b) Visual results of deviation optimization for spherical aggregation and linear aggregation under different aggregation steps.

Figure 5: Ablation study.

Table 4: Ablation results of different deviation optimization steps on COCO-MIG benchmark in InteractDiffusion(+MIGC).

| Deviation Optimization Step | Instance Success Rate (%) ↑ | | | | | | mIoU Score (%) ↑ | | | | | | CLIP Score ↑ | |
|---|---|---|---|---|---|---|---|---|---|---|---|---|---|---|
| | $L_2$ | $L_3$ | $L_4$ | $L_5$ | $L_6$ | Avg | $L_2$ | $L_3$ | $L_4$ | $L_5$ | $L_6$ | Avg | Global | Local |
| 0 | 61.56 | 52.08 | 53.75 | 51.62 | 51.87 | 53.18 | **54.76** | 45.09 | 46.36 | 45.48 | 45.79 | 46.62 | 31.09 | 27.56 |
| 0.1 | 57.60 | 53.26 | 53.89 | 52.40 | 52.08 | 53.26 | 53.27 | 45.53 | 46.86 | 46.31 | 45.80 | 46.83 | 31.43 | 27.87 |
| 0.2 | 59.62 | 54.27 | 54.06 | 51.25 | 52.29 | 53.40 | 52.71 | 46.32 | 47.17 | 44.68 | 45.74 | 46.59 | 31.92 | 28.45 |
| 0.3 | **62.29** | **54.33** | **56.31** | **53.87** | **52.89** | **54.78** | 54.30 | **46.37** | **48.44** | **47.65** | **46.64** | **47.74** | **32.81** | **28.96** |
| 0.4 | 61.34 | 53.62 | 55.32 | 52.22 | 52.50 | 53.96 | 53.46 | 46.21 | 46.82 | 45.26 | 46.23 | 46.78 | 32.23 | 28.55 |
| 0.5 | 58.30 | 53.43 | 55.14 | 51.76 | 51.76 | 53.27 | 52.88 | 45.03 | 46.52 | 44.95 | 45.51 | 46.14 | 31.15 | 27.78 |

bounding boxes or regions as additional conditions to enforce spatial constraints. More recently, several studies have pursued direct fine-grained multimodal control (Huang et al., 2023a; Smith et al., 2023; Gu et al., 2024; Kumari et al., 2023), but these approaches typically require extensive multi-condition datasets and increasingly complex model architectures.

**Compositional Generation.** Some recent works (Du et al., 2020; Liu et al., 2022; Du et al., 2023; Du & Kaelbling, 2024; Garipov et al., 2023; Bradley et al., 2025; Thornton et al., 2025; Skreta et al., 2025a; He et al., 2025; Skreta et al., 2025b) aim to create new distributions by combining different distributions, commonly using techniques such as linear weighted score, followed by simulated annealing or other optimization methods. Other works (Biggs et al., 2024; Oh et al., 2025; Wang et al., 2025) attempt to combine models by applying weighted interpolation to their parameters. However, this approach requires that the models share an identical architecture, which significantly restricts its applicability. A more in-depth comparison of our AMDM and its relation to compositional generation methods can be found in the Appendix E.

## 6 CONCLUSION

This paper proposes a novel AMDM algorithm designed for fine-grained generation, which consists of two main components: spherical aggregation and deviation optimization. Experimental results demonstrate the effectiveness of the AMDM algorithm, revealing that diffusion models initially prioritize image feature generation, shifting their focus to image quality and consistency in later stages. The algorithm provides a new perspective for addressing fine-grained generation challenges: We can leverage existing or develop new conditional diffusion models that control specific aspects, and then aggregate them using the AMDM algorithm. This eliminates the need for constructing complex datasets, designing intricate model architectures, and incurring high training costs.

## 7 REPRODUCIBILITY STATEMENT

We provide complete code and configurations in the anonymous supplementary materials (including evaluation scripts, environment specifications, dependency versions, and fixed random seeds) to enable one-click reproduction of our experiments. The required hardware and practical tips for reproduction are described in the Experiments section.

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

# Appendix

## A  CONCENTRATION OF MEASURE THEOREM FOR HIGH-DIMENSIONAL INDEPENDENT NORMAL DISTRIBUTION

**Lemma A.1.** *Let $X = (X_1, X_2, \ldots, X_n)$ be an $n$-dimensional random vector where each component $X_i$ is independently and identically distributed as $\mathcal{N}(0, \sigma^2)$. Then, as the dimension $n$ increases, the norm $\|X\|$ of $X$ is concentrated around $\sqrt{n}\sigma$. Specifically, for any small $\epsilon \geq 0$ and sufficiently large $n$,*

$$P\left(\left|\|X\| - \sqrt{n}\sigma\right| \leq \epsilon\sqrt{n}\sigma\right) \geq 1 - 2e^{-\frac{n\epsilon^2}{1+2\epsilon}}. \tag{9}$$

*This means $X$ will concentrate near the hypersphere of radius $\sqrt{n}\sigma$ with high probability.*

*Proof.* First, we consider the squared norm of a random vector $X : \|X\|^2 = X_1^2 + X_2^2 + \ldots + X_n^2$. Since $X_i \sim \mathcal{N}(0, \sigma^2)$, it follows that $Y = \|X\|^2/\sigma^2 \sim \chi^2(n)$.

Using the Chernoff bound of the chi-square distribution:

$$P(Y \geq (1+\delta)n) \leq e^{-\frac{n}{2}[\delta - \ln(1+\delta)]}, \tag{10}$$

$$P(Y \leq (1-\delta)n) \leq e^{-\frac{n}{2}[-\delta - \ln(1-\delta)]}, \tag{11}$$

where $\delta \geq 0$ is a small value. According to $\ln(1+\delta) \leq \frac{(2+\delta)\delta}{2(1+\delta)}$ and $\ln(1-\delta) \leq -\delta - \frac{\delta^2}{2}$, we can deduce that:

$$P(\|X\|^2 \geq (1+\delta)n\sigma^2) = P(Y \geq (1+\delta)n)$$
$$\leq e^{-\frac{n\delta^2}{4(1+\delta)}}, \tag{12}$$

$$P(\|X\|^2 \leq (1-\delta)n\sigma^2) = P(Y \leq (1-\delta)n)$$
$$\leq e^{-\frac{n\delta^2}{4}}$$
$$\leq e^{-\frac{n\delta^2}{4(1+\delta)}}. \tag{13}$$

Given the $\sqrt{1+\delta} \leq 1 + \frac{\delta}{2}$ and $\sqrt{1-\delta} \geq 1 - \frac{\delta}{2}$, we have:

$$P(\|X\| \geq (1 + \frac{\delta}{2})\sqrt{n}\sigma) \leq P(\|X\| \geq \sqrt{(1+\delta)}\sqrt{n}\sigma)$$
$$\leq e^{-\frac{n\delta^2}{4(1+\delta)}}, \tag{14}$$

$$P(\|X\| \leq (1 - \frac{\delta}{2})\sqrt{n}\sigma) \leq P(\|X\| \leq \sqrt{(1-\delta)}\sqrt{n}\sigma)$$
$$\leq e^{-\frac{n\delta^2}{4(1+\delta)}}. \tag{15}$$

Let $\epsilon = \delta/2$, and then organize equations (14) and (15):

$$P\left(\left|\|X\| - \sqrt{n}\sigma\right| \leq \epsilon\sqrt{n}\sigma\right) \geq 1 - 2e^{-\frac{n\epsilon^2}{1+2\epsilon}}, \tag{16}$$

which concludes the proof.

## B  PROOF OF PROPOSITION 3.1

**Proposition 3.1.** Assume that the aggregated variable at time step $t$ is $\mathbf{z}_t'$. For the sampling step from $t$ to $t-1$, two diffusion models $p_{\theta_1}$ and $p_{\theta_2}$ sample $\mathbf{z}_{t-1}^{\theta_1}$ and $\mathbf{z}_{t-1}^{\theta_2}$ respectively from (4). Then the approximate upper bound for the absolute value of the difference of the norms is:

$$\left|\|\mathbf{z}_{t-1}^{\theta_1}\| - \|\mathbf{z}_{t-1}^{\theta_2}\|\right| \leq \sqrt{1 - 2\alpha(1 - \cos\varphi)}\delta, \quad \cos\varphi \geq 1 - \frac{\delta^2}{\|\mathbf{z}_{t-1}^{\theta_1}\|^2 + \|\mathbf{z}_{t-1}^{\theta_2}\|^2}, \tag{5}$$

where $\varphi$ is the angle between $\mathbf{z}_{t-1}^{\theta_1}$ and $\mathbf{z}_{t-1}^{\theta_2}$, $\alpha \in [0, 1/4]$, $\delta \leq B_t(L_{\theta_1}L_{\mathcal{T}} + L_t) + \sigma_t\sqrt{2n}$ is the Lipschitz constant, and $B_t$ is the coefficient of $\epsilon$ in the sampling process (4).

*Proof.* To begin with, by applying the Cauchy–Schwarz inequality, we have:

$$
\begin{aligned}
\left| \|\mathbf{z}_{t-1}^{\theta_1}\| - \|\mathbf{z}_{t-1}^{\theta_2}\| \right| &= \frac{\left| \|\mathbf{z}_{t-1}^{\theta_1}\|^2 - \|\mathbf{z}_{t-1}^{\theta_2}\|^2 \right|}{\|\mathbf{z}_{t-1}^{\theta_1}\| + \|\mathbf{z}_{t-1}^{\theta_2}\|} \\
&= \frac{\left| \langle \mathbf{z}_{t-1}^{\theta_1} + \mathbf{z}_{t-1}^{\theta_2}, \ \mathbf{z}_{t-1}^{\theta_1} - \mathbf{z}_{t-1}^{\theta_2} \rangle \right|}{\|\mathbf{z}_{t-1}^{\theta_1}\| + \|\mathbf{z}_{t-1}^{\theta_2}\|} \\
&\leq \frac{\|\mathbf{z}_{t-1}^{\theta_1} + \mathbf{z}_{t-1}^{\theta_2}\|}{\|\mathbf{z}_{t-1}^{\theta_1}\| + \|\mathbf{z}_{t-1}^{\theta_2}\|} \|\mathbf{z}_{t-1}^{\theta_1} - \mathbf{z}_{t-1}^{\theta_2}\|.
\end{aligned}
\tag{17}
$$

For the term on the left-hand side, we have:

$$
\begin{aligned}
\frac{\|\mathbf{z}_{t-1}^{\theta_1} + \mathbf{z}_{t-1}^{\theta_2}\|}{\|\mathbf{z}_{t-1}^{\theta_1}\| + \|\mathbf{z}_{t-1}^{\theta_2}\|} &= \sqrt{\left( \frac{\|\mathbf{z}_{t-1}^{\theta_1} + \mathbf{z}_{t-1}^{\theta_2}\|}{\|\mathbf{z}_{t-1}^{\theta_1}\| + \|\mathbf{z}_{t-1}^{\theta_2}\|} \right)^2} \\
&= \sqrt{\frac{(\|\mathbf{z}_{t-1}^{\theta_1}\| + \|\mathbf{z}_{t-1}^{\theta_2}\|)^2 - 2\|\mathbf{z}_{t-1}^{\theta_1}\|\|\mathbf{z}_{t-1}^{\theta_2}\|(1 - \cos\varphi)}{(\|\mathbf{z}_{t-1}^{\theta_1}\| + \|\mathbf{z}_{t-1}^{\theta_2}\|)^2}} \\
&= \sqrt{1 - \frac{2\|\mathbf{z}_{t-1}^{\theta_1}\|\|\mathbf{z}_{t-1}^{\theta_2}\|}{(\|\mathbf{z}_{t-1}^{\theta_1}\| + \|\mathbf{z}_{t-1}^{\theta_2}\|)^2}(1 - \cos\varphi)} \\
&= \sqrt{1 - 2\alpha(1 - \cos\varphi)},
\end{aligned}
\tag{18}
$$

where $\alpha = \dfrac{\|\mathbf{z}_{t-1}^{\theta_1}\| \|\mathbf{z}_{t-1}^{\theta_2}\|}{\left( \|\mathbf{z}_{t-1}^{\theta_1}\| + \|\mathbf{z}_{t-1}^{\theta_2}\| \right)^2}$. By the basic inequality, it follows that $\alpha \in [0, \frac{1}{4}]$, and moreover $1 - 2\alpha(1 - \cos\varphi) \leq 1$. For the term on the right-hand side, we denote $\delta = \|\mathbf{z}_{t-1}^{\theta_1} - \mathbf{z}_{t-1}^{\theta_2}\|$ and abbreviate it as:

$$
\begin{aligned}
\mathbf{z}_{t-1}^{\theta_1} &= A_t \mathbf{z}_t' + B_t \boldsymbol{\epsilon}_{\theta_1}(\mathbf{z}_t', t, y_1) + \sigma_t \epsilon_1, \\
\mathbf{z}_{t-1}^{\theta_2} &= A_t \mathbf{z}_t' + B_t \boldsymbol{\epsilon}_{\theta_2}(\mathbf{z}_t', t, y_2) + \sigma_t \epsilon_2.
\end{aligned}
\tag{19}
$$

Thus:

$$
\begin{aligned}
\delta &= \|\mathbf{z}_{t-1}^{\theta_1} - \mathbf{z}_{t-1}^{\theta_2}\| \\
&= \|A_t \mathbf{z}_t' + B_t \boldsymbol{\epsilon}_{\theta_1}(\mathbf{z}_t', t, y_1) + \sigma_t \epsilon_1 - (A_t \mathbf{z}_t' + B_t \boldsymbol{\epsilon}_{\theta_2}(\mathbf{z}_t', t, y_2) + \sigma_t \epsilon_2)\| \\
&= \|B_t(\boldsymbol{\epsilon}_{\theta_1}(\mathbf{z}_t', t, y_1) - \boldsymbol{\epsilon}_{\theta_2}(\mathbf{z}_t', t, y_2)) + \sigma_t(\epsilon_1 - \epsilon_2)\| \\
&\leq B_t \|\boldsymbol{\epsilon}_{\theta_1}(\mathbf{z}_t', t, y_1) - \boldsymbol{\epsilon}_{\theta_2}(\mathbf{z}_t', t, y_2)\| + \sigma_t \|\epsilon_1 - \epsilon_2\|.
\end{aligned}
\tag{20}
$$

For the first term of (20):

$$
\begin{aligned}
&B_t \|\boldsymbol{\epsilon}_{\theta_1}(\mathbf{z}_t', t, y_1) - \boldsymbol{\epsilon}_{\theta_2}(\mathbf{z}_t', t, y_2)\| \\
=&B_t \|\boldsymbol{\epsilon}_{\theta_1}(\mathbf{z}_t', t, y_1) - \boldsymbol{\epsilon}_{\theta_1}(\mathbf{z}_t', t, y_2) + \boldsymbol{\epsilon}_{\theta_1}(\mathbf{z}_t', t, y_2) - \boldsymbol{\epsilon}_{\theta_2}(\mathbf{z}_t', t, y_2)\| \\
\leq&B_t(\|\boldsymbol{\epsilon}_{\theta_1}(\mathbf{z}_t', t, y_1) - \boldsymbol{\epsilon}_{\theta_1}(\mathbf{z}_t', t, y_2)\| + \|\boldsymbol{\epsilon}_{\theta_1}(\mathbf{z}_t', t, y_2) - \boldsymbol{\epsilon}_{\theta_2}(\mathbf{z}_t', t, y_2)\|).
\end{aligned}
\tag{21}
$$

Due to the presence of dot-product self-attention and layer normalization, the U-Net architecture with Transformer modules in diffusion models cannot satisfy global Lipschitz continuity (Kim et al., 2021; Castin et al., 2024; Qi et al., 2023). However, note that in $\|\boldsymbol{\epsilon}_{\theta_1}(\mathbf{z}_t', t, y_1) - \boldsymbol{\epsilon}_{\theta_1}(\mathbf{z}_t', t, y_2)\|$, the conditions $y_1$ and $y_2$ belong to the same fine-grained task setting. Therefore, within this setting, the U-Net architecture admits a local Lipschitz bound $L_{\theta_1}$ with respect to the conditioning variable. Hence:

$$
\begin{aligned}
&B_t \|\boldsymbol{\epsilon}_{\theta_1}(\mathbf{z}_t', t, y_1) - \boldsymbol{\epsilon}_{\theta_2}(\mathbf{z}_t', t, y_2)\| \\
\leq&B_t(\|\boldsymbol{\epsilon}_{\theta_1}(\mathbf{z}_t', t, y_1) - \boldsymbol{\epsilon}_{\theta_1}(\mathbf{z}_t', t, y_2)\| + \|\boldsymbol{\epsilon}_{\theta_1}(\mathbf{z}_t', t, y_2) - \boldsymbol{\epsilon}_{\theta_2}(\mathbf{z}_t', t, y_2)\|) \\
\leq&B_t(L_{\theta_1} \|y_1 - y_2\| + \|\boldsymbol{\epsilon}_{\theta_1}(\mathbf{z}_t', t, y_2) - \boldsymbol{\epsilon}_{\theta_2}(\mathbf{z}_t', t, y_2)\|) \\
\leq&B_t(L_{\theta_1} L_{\mathcal{T}} + L_t).
\end{aligned}
\tag{22}
$$

For the second term of (20), since $\epsilon_1$ and $\epsilon_2$ are independent Gaussian noises, we have $\sigma_t \| \epsilon_1 - \epsilon_2 \| \approx \sqrt{2n}$. Therefore:

$$
\begin{aligned}
\delta &\leq B_t \| \boldsymbol{\epsilon}_{\theta_1}(\mathbf{z}_t', t, y_1) - \boldsymbol{\epsilon}_{\theta_2}(\mathbf{z}_t', t, y_2) \| + \sigma_t \| \epsilon_1 - \epsilon_2 \| \\
&\leq B_t (L_{\theta_1} L_{\mathcal{T}} + L_t) + \sigma_t \sqrt{2n}.
\end{aligned}
\tag{23}
$$

Combining with (18), we finally obtain:

$$
\begin{aligned}
\left| \| \mathbf{z}_{t-1}^{\theta_1} \| - \| \mathbf{z}_{t-1}^{\theta_2} \| \right| &\leq \frac{\| \mathbf{z}_{t-1}^{\theta_1} + \mathbf{z}_{t-1}^{\theta_2} \|}{\| \mathbf{z}_{t-1}^{\theta_1} \| + \| \mathbf{z}_{t-1}^{\theta_2} \|} \| \mathbf{z}_{t-1}^{\theta_1} - \mathbf{z}_{t-1}^{\theta_2} \| \\
&\leq \sqrt{1 - 2\alpha(1 - \cos\varphi)} \delta.
\end{aligned}
\tag{24}
$$

Additionally,

$$
\begin{aligned}
\cos\varphi &= \frac{\| \mathbf{z}_{t-1}^{\theta_1} \|^2 + \| \mathbf{z}_{t-1}^{\theta_2} \|^2 - \delta^2}{2\| \mathbf{z}_{t-1}^{\theta_1} \| \| \mathbf{z}_{t-1}^{\theta_2} \|} \\
&\geq \frac{\| \mathbf{z}_{t-1}^{\theta_1} \|^2 + \| \mathbf{z}_{t-1}^{\theta_2} \|^2 - \delta^2}{\| \mathbf{z}_{t-1}^{\theta_1} \|^2 + \| \mathbf{z}_{t-1}^{\theta_2} \|^2} \\
&= 1 - \frac{\delta^2}{\| \mathbf{z}_{t-1}^{\theta_1} \|^2 + \| \mathbf{z}_{t-1}^{\theta_2} \|^2},
\end{aligned}
\tag{25}
$$

which concludes the proof.

## C  PROOF OF PROPOSITION 3.2

**Proposition 3.2.** For the diffusion model $p_{\theta_1}$ defined by (4) and any new intermediate variable $\mathbf{z}_{t-1}'$ from (6), let:

$$
\tilde{\mathbf{z}}_{t-1} = \mathbf{z}_{t-1}' - \eta_{t-1}^{\theta_1} \frac{\mathbf{z}_{t-1}' - \mu_{\theta_1}(\mathbf{z}_t^{\theta_1}, t, y_1)}{\| \mathbf{z}_{t-1}' - \mu_{\theta_1}(\mathbf{z}_t^{\theta_1}, t, y_1) \|},
\tag{7}
$$

where $\eta_{t-1}^{\theta_1}$ is a small optimization step size. There exists $\eta_{t-1}^{\theta_1}$ such that $\tilde{\mathbf{z}}_{t-1} \in D_{t-1,y_1}^{\theta_1}$. Moreover, an approximate lower bound on the probability that $\tilde{\mathbf{z}}_{t-1} \in D_{t-1,y_2}^{\theta_2}$ is given by:

$$
P\left( \tilde{\mathbf{z}}_{t-1} \in D_{t-1,y_2}^{\theta_2} \right) \geq 1 - 2\exp\left( -\frac{n\left( \epsilon_{t-1}^{\theta_2} - \frac{d}{\sigma_t \sqrt{n}} \right)^2}{1 + 2\left( \epsilon_{t-1}^{\theta_2} - \frac{d}{\sigma_t \sqrt{n}} \right)} \right),
\tag{8}
$$

where $d = \phi_w(\varphi) \| \mathbf{z}_{t-1}^{\theta_1} - \mathbf{z}_{t-1}^{\theta_2} \| + \eta_{t-1}^{\theta_1}$ and $\phi_w(\varphi) = \sin\big((1 - w)\varphi/2\big) / \sin(\varphi/2)$.

*Proof.*

(1) The existence of $\eta_{t-1}^{\theta_1}$ such that $\tilde{\mathbf{z}}_{t-1} \in D_{t-1,y_1}^{\theta_1}$:

We begin by defining the spherical shell as the high-probability set of $p_{\theta_i}(\mathbf{z}_{t-1}^{\theta_i} \mid \mathbf{z}_t^{\theta_i}, y_i)$:

$$
\mathcal{A}_{t-1,y_i}^{\theta_i}(\epsilon_{t-1}^{\theta_i}) = \left\{ \left| \| \mathbf{z}_{t-1}^{\theta_i} - \mu_{\theta_i}(\mathbf{z}_t^{\theta_i}, t, y_i) \| - \sqrt{n}\sigma_t \right| \leq \epsilon_{t-1}^{\theta_i} \sqrt{n}\sigma_t \right\},
\tag{26}
$$

where $\epsilon_{t-1}^{\theta_i} = \sup\left\{ \epsilon \geq 0 : \mathcal{A}_{t-1,y_i}^{\theta_i}(\epsilon) \subseteq D_{t-1,y_i}^{\theta_i}(\tau_{t-1}^{\theta_i}) \right\}$.

Note that:

$$
\begin{aligned}
\nabla_{\mathbf{z}_{t-1}'} p_{\theta_1}(\mathbf{z}_{t-1}' | \mathbf{z}_t^{\theta_1}, y_1) &= \frac{1}{(2\pi\sigma_t^2)^{n/2}} \nabla_{\mathbf{z}_{t-1}'} \left( e^{-\frac{\| \mathbf{z}_{t-1}' - \mu_\theta(\mathbf{z}_t^{\theta_1}, t, y_1) \|^2}{2\sigma_t^2}} \right) \\
&= -\frac{1}{(2\pi\sigma_t^2)^{n/2}} e^{-\frac{\| \mathbf{z}_{t-1}' - \mu_\theta(\mathbf{z}_t^{\theta_1}, t, y_1) \|^2}{2\sigma_t^2}} \left( \frac{\mathbf{z}_{t-1}' - \mu_\theta(\mathbf{z}_t^{\theta_1}, t, y_1)}{\sigma_t^2} \right).
\end{aligned}
\tag{27}
$$

Therefore,

$$
\begin{aligned}
\tilde{\mathbf{z}}_{t-1} &= \mathbf{z}'_{t-1} - \eta_{t-1}^{\theta_1} \frac{\mathbf{z}'_{t-1} - \mu_\theta(\mathbf{z}_t^{\theta_1}, t, y_1)}{\|\mathbf{z}'_{t-1} - \mu_\theta(\mathbf{z}_t^{\theta_1}, t, y_1)\|} \\
&= \mathbf{z}'_{t-1} + \eta_{t-1}^{\theta_1} \frac{\nabla_{\mathbf{z}'_{t-1}} p_{\theta_1}(\mathbf{z}_{t-1}' | \mathbf{z}_t^{\theta_1}, y_1)}{\|\nabla_{\mathbf{z}'_{t-1}} p_{\theta_1}(\mathbf{z}_{t-1}' | \mathbf{z}_t^{\theta_1}, y_1)\|}.
\end{aligned}
\tag{28}
$$

This indicates that $\tilde{\mathbf{z}}_{t-1}$ is obtained from $\mathbf{z}'_{t-1}$ by moving $\eta_{t-1}^{\theta_1}$ units in the radial direction opposite to $p_{\theta_1}(\mathbf{z}'_{t-1} \mid \mathbf{z}_t^{\theta_1}, y_1)$. Then, it suffices to ensure that:

$$
\left| \|\mathbf{z}'_{t-1} - \mu_\theta(\mathbf{z}_t^{\theta_1}, t, y_1)\| + \eta_{t-1}^{\theta_1} - \sqrt{n}\,\sigma_t \right| < \epsilon_{t-1}^{\theta_1} \sqrt{n}\,\sigma_t,
\tag{29}
$$

from which we obtain:

$$
\begin{aligned}
&-\epsilon_{t-1}^{\theta_1} \sqrt{n}\sigma_t - \left\|\mathbf{z}'_{t-1} - \mu_\theta(\mathbf{z}_t^{\theta_1}, t, y_1)\right\| + \sqrt{n}\sigma_t \\
&\quad < \eta_{t-1}^{\theta_1} < \\
&\epsilon_{t-1}^{\theta_1} \sqrt{n}\sigma_t - \left\|\mathbf{z}'_{t-1} - \mu_\theta(\mathbf{z}_t^{\theta_1}, t, y_1)\right\| + \sqrt{n}\sigma_t,
\end{aligned}
\tag{30}
$$

thus, we have $\tilde{\mathbf{z}}_{t-1} \in \mathcal{A}_{t-1,y_1}^{\theta_1} \subseteq D_{t-1,y_1}^{\theta_1}$, which completes the proof of existence.

(2) The approximate lower bound on the probability that $\tilde{\mathbf{z}}_{t-1} \in D_{t-1,y_2}^{\theta_2}$:

By the triangle inequality, for any $z_{t-1}^{\theta_2} \in \mathcal{A}_{t-1,y_2}^{\theta_2}$, we have:

$$
\left\|\tilde{\mathbf{z}}_{t-1} - \mathbf{z}_{t-1}^{\theta_2}\right\| \le \left\|\mathbf{z}'_{t-1} - \mathbf{z}_{t-1}^{\theta_2}\right\| + \eta_{t-1}^{\theta_1}.
\tag{31}
$$

From proposition 3.1, we assume that $\left\|\mathbf{z}_{t-1}^{\theta_1}\right\| \approx \left\|\mathbf{z}_{t-1}^{\theta_2}\right\| \approx r_{t-1}$, then, the chord lengths satisfy:

$$
\left\|\mathbf{z}'_{t-1} - \mathbf{z}_{t-1}^{\theta_2}\right\| = 2r_{t-1}\sin\left(\frac{(1-w)\varphi}{2}\right), \qquad \delta = \left\|\mathbf{z}_{t-1}^{\theta_1} - \mathbf{z}_{t-1}^{\theta_2}\right\| = 2r_{t-1}\sin\left(\frac{\varphi}{2}\right).
\tag{32}
$$

Dividing the two identities yields:

$$
\left\|\mathbf{z}'_{t-1} - \mathbf{z}_{t-1}^{\theta_2}\right\| = \phi_w(\varphi)\,\delta, \quad \phi_w(\varphi) = \frac{\sin\left(\frac{(1-w)\varphi}{2}\right)}{\sin\left(\frac{\varphi}{2}\right)}.
\tag{33}
$$

Combining (31) and (33), denote $d = \phi_w(\varphi)\,\delta + \eta_{t-1}^{\theta_1}$ and We can obtain the bound:

$$
\mathrm{dist}\left(\tilde{\mathbf{z}}_{t-1}, \mathcal{A}_{t-1,y_2}^{\theta_2}\right) \le d.
\tag{34}
$$

So, we can also refer to $d$ as the maximum distance to $p_{\theta_2}$. This implies that, if:

$$
\left| \|\mathbf{z}_{t-1}^{\theta_2} - \mu_{\theta_2}\| - \sigma_t\sqrt{n} \right| \le (\epsilon_{t-1}^{\theta_2} - d/\sigma_t\sqrt{n})\sigma_t\sqrt{n},
\tag{35}
$$

then $B\left(\mathbf{z}_{t-1}^{\theta_2}, d\right) \subseteq \mathcal{A}_{t-1,y_2}^{\theta_2}$. Therefore:

$$
\begin{aligned}
P\left(\tilde{\mathbf{z}}_{t-1} \in D_{t-1,y_2}^{\theta_2}\right) &\ge P\left(\tilde{\mathbf{z}}_{t-1} \in \mathcal{A}_{t-1,y_2}^{\theta_2}\right) \\
&\ge P\left(\left| \|\mathbf{z}_{t-1}^{\theta_2} - \mu_{\theta_2}\| - \sigma_t\sqrt{n} \right| \le \left(\epsilon_{t-1}^{\theta_2} - \tfrac{d}{\sigma_t\sqrt{n}}\right)\sigma_t\sqrt{n}\right) \\
&\ge 1 - 2\exp\left(-\frac{n\left(\epsilon_{t-1}^{\theta_2} - \frac{d}{\sigma_t\sqrt{n}}\right)^2}{1 + 2\left(\epsilon_{t-1}^{\theta_2} - \frac{d}{\sigma_t\sqrt{n}}\right)}\right).
\end{aligned}
\tag{36}
$$

The first line follows from $\tilde{\mathbf{z}}_{t-1} \in \mathcal{A}_{t-1,y_2}^{\theta_2} \subseteq D_{t-1,y_2}^{\theta_2}$, and the third line follows from Lemma A.1, which concludes the proof.

# D  STATISTICAL ANALYSIS

## D.1  GAUSSIANITY IN LATENT SPACE

Chung et al. (2022) have shown that $\mathcal{M}_t$ is concentrated on an $(n-1)$-dimensional manifold, which approximates an $n$-dimensional hypersphere as $t$ becomes large. Motivated by this property and proposition 3.1, the AMDM algorithm performs spherical aggregation in order to reduce manifold deviation. Next, we focus on the statistical characteristics of the distribution $p(\mathbf{z}_t)$, in particular its first and second-order moments, to validate the appropriateness of the hyperspherical approximation in the large $t$ regime.

The continuous form of (1) can be rewritten as:

$$d\mathbf{z}_t = -\frac{1}{2}g_t^2 \mathbf{z}_t dt + g_t d\mathbf{w}_t, \tag{37}$$

We can use equations (5.50) and (5.51) (Särkkä & Solin, 2019) to derive the relationship between the mean and variance of (37) as they evolve over time:

$$\frac{d\mathbf{m}}{dt} = \mathbb{E}\left[-\frac{1}{2}g_t^2 \mathbf{z}_t\right] = -\frac{1}{2}g_t^2 \mathbf{m}, \tag{38}$$

$$\frac{d\mathbf{P}}{dt} = \mathbb{E}\left[-\frac{1}{2}g_t^2 \mathbf{z}_t(\mathbf{z}_t - \mathbf{m})^T\right] + \mathbb{E}\left[-\frac{1}{2}g_t^2(\mathbf{z}_t - \mathbf{m})\mathbf{z}_t^T\right] + g_t^2 \boldsymbol{I}$$
$$= -g_t^2 \mathbf{P} + g_t^2 \boldsymbol{I}. \tag{39}$$

The solutions are:

$$\mathbf{m}(t) = \mathbf{m}(0)e^{-\frac{1}{2}\int_0^t g_s^2 ds} \tag{40}$$

$$\mathbf{P}(t) = \boldsymbol{I} + (\mathbf{P}(0) - \boldsymbol{I})\,e^{-\int_0^t g_s^2 ds}, \tag{41}$$

where $g_t = \sqrt{\beta_t}$, and we set $\beta_t$ to follow the linear schedule in SD (Rombach et al., 2022). For $\mathbf{m}(0)$ and $\mathbf{P}(0)$, since the data $\mathbf{z}_0^{(i)} \in [-1, 1]$, it follows that $\mathbf{m}(0)^{(i)} \in [-1, 1]$ and $\mathbf{P}(0)^{(i)} \in [0, 1]$. Therefore, we consider the extreme case with $\mathbf{m}(0) = \mathbf{1}$ and $\mathbf{P}(0) = \mathbf{0}$. The resulting mean and variance of $p(\mathbf{z}_t)$ as functions of time are shown in Figure 6.

From Figure 6, it can be observed that when $t > 0.6$, the mean rapidly converges to 0 while the variance rapidly converges to 1, approaching a standard Gaussian distribution. When the Gaussianity of $\mathcal{M}_t$ is well-behaved, implying that the global geometry exhibits desirable spherical characteristics, it is both reasonable and natural to regard two vectors with a small angular separation and equal magnitudes as lying on a local sphere.

## D.2  NUMERICAL ANALYSIS OF THE EXPERIMENTS FOR PROPOSITION 3.1

To assess the validity of Proposition 3.1, we log intermediate variables from InteractDiffusion(+MIGC) and report them in Table 5 and Figure 7. During the first $s = 20$ aggregation steps ($t = 50$ to $t = 30$), $\left|\|\mathbf{z}_t^{\theta_1}\| - \|\mathbf{z}_t^{\theta_2}\|\right| \approx 0$ and $\varphi$ is small, which directly supports the analysis of Proposition 3.1. This behavior is expected for the following reasons. At initialization ($t = 50$), both trajectories are sampled from the standard Gaussian and thus have numerically almost identical magnitudes; the subsequent single-step denoising from $t = 50 \to 49$, although performed by different models, introduces only a small variations, so the magnitudes remain nearly equal. Moreover, the small variations in the angle $\varphi$ and in the difference norm $\delta = \|\mathbf{z}_t^{\theta_1} - \mathbf{z}_t^{\theta_2}\|$ further indicate that the per-step generative error is well controlled. Once aggregation takes effect, $\mathbf{z}_{t-1}^{\theta_1}$ and $\mathbf{z}_{t-1}^{\theta_2}$ start from the common aggregation point $\mathbf{z}_t'$, undergo small deviation optimization, and are then sampled separately; because the aggregation point is shared, the deviations are limited, and each sampling step introduces only a small error (conditional proximity and functional proximity), the magnitudes remain almost equal. Taken together, these observations justify the equal-norm assumption throughout the aggregation process.

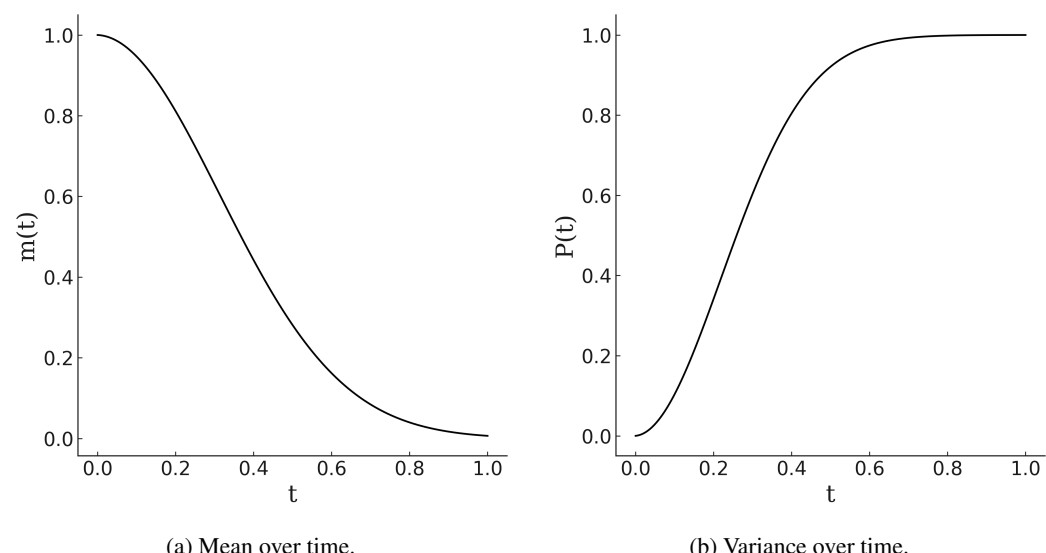

(a) Mean over time.

(b) Variance over time.

Figure 6: Statistical properties of $p(\mathbf{z}_t)$ over time.

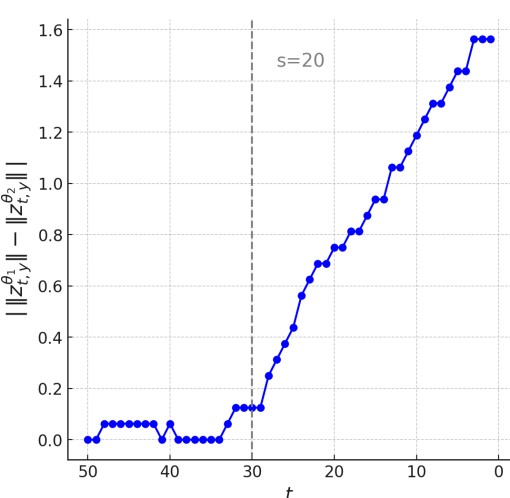

Figure 7: Difference of norm varying with time.

### D.3 MAXIMUM DISTANCE TERM

In Proposition 3.2, we derived the approximate lower bound of the probability:

$$P\left(\tilde{\mathbf{z}}_{t-1} \in D_{t-1,y_2}^{\theta_2}\right) \geq 1 - 2\exp\left(-\frac{n\left(\epsilon_{t-1}^{\theta_2} - \frac{d}{\sigma_t\sqrt{n}}\right)^2}{1 + 2\left(\epsilon_{t-1}^{\theta_2} - \frac{d}{\sigma_t\sqrt{n}}\right)}\right), \tag{8}$$

where the key quantity is the maximum distance to the model, given by $d = \phi_w(\varphi)\,\delta + \eta_{t-1}^{\theta_1}$. The numerical value of $d$ directly determines whether the optimized variable $\mathbf{z}_t^{\theta_1}$ is likely to lie on the data manifold of model $\theta_2$, thereby enabling fine-grained generation. Theoretically, the smaller the value of $d$, the higher the likelihood. To this end, we separately analyze $\phi_w(\varphi) = \frac{\sin\left((1-w)\varphi/2\right)}{\sin(\varphi/2)}$ and $\delta = \left\|\mathbf{z}_{t-1}^{\theta_1} - \mathbf{z}_{t-1}^{\theta_2}\right\|$.

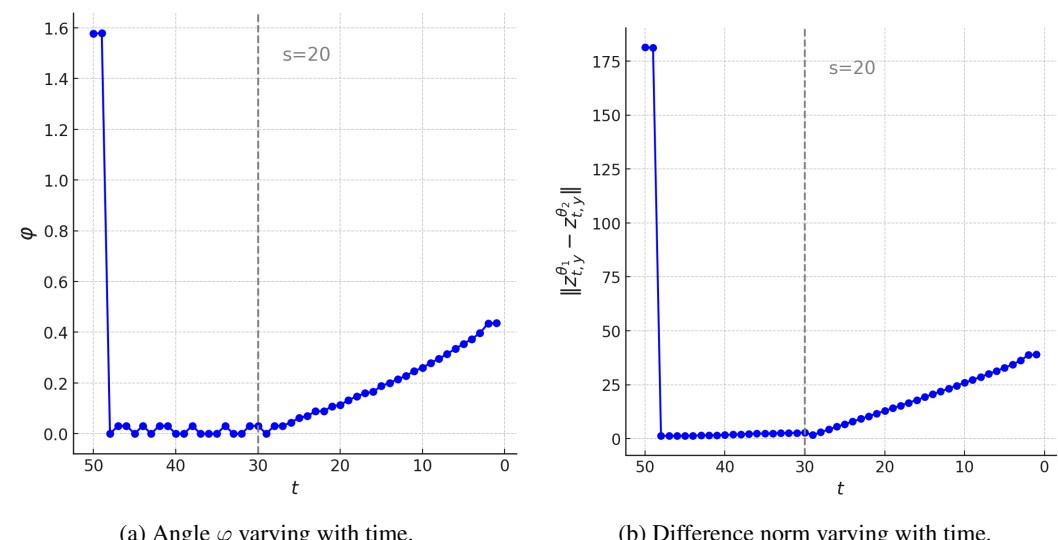

(a) Angle $\varphi$ varying with time.  (b) Difference norm varying with time.

Figure 8: Variation of angle and the difference norm.

For $\phi_w(\varphi)$, we first examine the variation of the angle $\varphi$ over time. The detailed experimental results can be referred to in the Table 5 and Fig 8a. At the initial step $t = 50$, since random points in high-dimensional Gaussian space are almost orthogonal, the angle satisfies $\varphi \approx \frac{\pi}{2}$. At $t = 49$, a single sampling step of the model introduces only a negligible perturbation, so $\varphi$ remains essentially unchanged. Subsequently, following an argument similar to the analysis of difference of norm in Appendix D.2: since $\mathbf{z}_{t-1}^{\theta_1}$ and $\mathbf{z}_{t-1}^{\theta_2}$ are both derived from the common aggregation point $\mathbf{z}_t'$, and then undergo deviation optimization and independent sampling, these operations only introduce small perturbations, implying that $\varphi$ also stays small. Expanding $\phi_w(\varphi)$ in a Taylor series gives $\phi_w(\varphi) = (1-w) + O(\varphi^2)$. Since $\varphi$ is very small, the higher-order terms can be neglected, yielding $\phi_w(\varphi) \approx (1-w)$. In summary, at the initial step, $\phi_w(\varphi) \approx \phi_w\left(\frac{\pi}{2}\right) = \sqrt{2}\sin\left(\frac{(1-w)\pi}{4}\right) \leq 1$, whereas after aggregation, $\phi_w(\varphi) \approx (1-w) \leq 1$. Thus, we conclude that $\phi_w(\varphi) \leq 1$.

For $\delta$, the specific experimental values are provided in the Table 5 and Figure 8b. Based on a similar analysis as above, it can be inferred that at the initial steps $t = 50$ and $t = 49$, the vectors are generated by random sampling, leading to a large magnitude of their difference. In subsequent steps, however, since both vectors originate from the same aggregation point of the previous step and the following operations introduce only small perturbations, the magnitude of their difference becomes significantly smaller.

In summary, only after the first aggregation optimization may the variable $\tilde{\mathbf{z}}_{T-1}$ deviate from $D_{T-1,y_2}^{\theta_2}$. In the subsequent aggregation steps, however, $\tilde{\mathbf{z}}_{t,y_2}$ remains, with high probability, within $D_{t,y_2}^{\theta_2}$, thereby fully incorporating the characteristics of different models and enabling fine-grained generation. This observation is also intuitive: since the initial sampling is performed from two random points that are relatively far apart, it is difficult for the first aggregation to lie simultaneously on the manifolds of both models at step $T - 1$.

In addition, we conducted 10 additional tests with different random seeds. The results show that the angle $\varphi$ and the difference of norm $\left| \|\mathbf{z}_t^{\theta_1}\| - \|\mathbf{z}_t^{\theta_2}\| \right|$ remain nearly identical across experiments, while the difference norm $\left\| \mathbf{z}_{t-1}^{\theta_1} - \mathbf{z}_{t-1}^{\theta_2} \right\|$ fluctuates only within $\pm 1$. Hence, the above analysis can be regarded as robust.

Table 5: Values of different intermediate variables over time in InteractDiffusion (+MIGC)

| $t$ | $\varphi$ | $\left\|\mathbf{z}_t^{\theta_1} - \mathbf{z}_t^{\theta_2}\right\|$ | $\left\|\left\|\mathbf{z}_t^{\theta_1}\right\| - \left\|\mathbf{z}_t^{\theta_2}\right\|\right\|$ |
|---|---|---|---|
| 50 | 1.5781 | 181.3750 | 0.0000 |
| 49 | 1.5791 | 181.2500 | 0.0000 |
| 48 | 0.0000 | 1.3086 | 0.0625 |
| 47 | 0.0312 | 1.3125 | 0.0625 |
| 46 | 0.0312 | 1.3281 | 0.0625 |
| 45 | 0.0000 | 1.3652 | 0.0625 |
| 44 | 0.0312 | 1.4219 | 0.0625 |
| 43 | 0.0000 | 1.5244 | 0.0625 |
| 42 | 0.0312 | 1.5625 | 0.0625 |
| 41 | 0.0312 | 1.6426 | 0.0000 |
| 40 | 0.0000 | 1.7812 | 0.0625 |
| 39 | 0.0000 | 1.8926 | 0.0000 |
| 38 | 0.0312 | 2.0586 | 0.0000 |
| 37 | 0.0000 | 2.2402 | 0.0000 |
| 36 | 0.0000 | 2.3730 | 0.0000 |
| 35 | 0.0000 | 2.4258 | 0.0000 |
| 34 | 0.0312 | 2.4707 | 0.0000 |
| 33 | 0.0000 | 2.5508 | 0.0625 |
| 32 | 0.0000 | 2.6309 | 0.1250 |
| 31 | 0.0312 | 2.7324 | 0.1250 |
| 30 | 0.0312 | 2.8184 | 0.1250 |
| 29 | 0.0000 | 1.6904 | 0.1250 |
| 28 | 0.0312 | 2.9727 | 0.2500 |
| 27 | 0.0312 | 4.2734 | 0.3125 |
| 26 | 0.0442 | 5.5273 | 0.3750 |
| 25 | 0.0625 | 6.7500 | 0.4375 |
| 24 | 0.0699 | 7.9648 | 0.5625 |
| 23 | 0.0884 | 9.1797 | 0.6250 |
| 22 | 0.0884 | 10.4141 | 0.6875 |
| 21 | 0.1083 | 11.6406 | 0.6875 |
| 20 | 0.1127 | 12.8750 | 0.7500 |
| 19 | 0.1327 | 14.1250 | 0.7500 |
| 18 | 0.1467 | 15.3750 | 0.8125 |
| 17 | 0.1595 | 16.6406 | 0.8125 |
| 16 | 0.1655 | 17.9219 | 0.8750 |
| 15 | 0.1877 | 19.2500 | 0.9375 |
| 14 | 0.2004 | 20.5938 | 0.9375 |
| 13 | 0.2146 | 21.9219 | 1.0625 |
| 12 | 0.2280 | 23.2500 | 1.0625 |
| 11 | 0.2467 | 24.5781 | 1.1250 |
| 10 | 0.2603 | 25.9219 | 1.1875 |
| 9 | 0.2786 | 27.2656 | 1.2500 |
| 8 | 0.2959 | 28.6094 | 1.3125 |
| 7 | 0.3137 | 29.9844 | 1.3125 |
| 6 | 0.3352 | 31.3906 | 1.3750 |
| 5 | 0.3540 | 32.8438 | 1.4375 |
| 4 | 0.3733 | 34.4062 | 1.4375 |
| 3 | 0.3967 | 36.1875 | 1.5625 |
| 2 | 0.4353 | 38.8125 | 1.5625 |
| 1 | 0.4365 | 38.9688 | 1.5625 |

# E COMPARISON WITH COMPOSITIONAL METHODS

Compositional generation aims to synthesize complex new samples that simultaneously satisfy multiple attributes or concepts by combining information from multiple different distributions. In the context of diffusion models, this is often achieved by combining the score functions $\nabla_x \log p_t(x)$ of the noise processes from different models to simulate a new reverse stochastic differential equation (SDE).

Du et al. (2023) proposed an energy-based framework for compositional generation, where the scores from multiple models are summed to approximate the score of the joint distribution, that is:

$$\nabla \log p_{\text{PoE}}(x) \approx \sum_{i=1}^{N} \nabla \log p_i(x),$$

and the corresponding target distribution is the Product of Experts (PoE):

$$p_{\text{PoE}}(x) \propto \prod_{i=1}^{N} p_i(x).$$

Sampling is performed using MCMC methods such as Langevin dynamics or HMC. However, these approaches are heuristic in nature, lack theoretical guarantees, and incur high computational costs.

Skreta et al. (2025a) introduce the *Feynman-Kac Corrector (FKC)* framework, which provides a principled approach to sampling from modified target distributions:

$$\textbf{Annealed:} \quad p_{t,\beta}^{\text{anneal}}(x) = \frac{1}{Z_t(\beta)} q_t(x)^{\beta},$$

$$\textbf{Product:} \quad p_t^{\text{prod}}(x) = \frac{1}{Z_t} q_t^1(x) q_t^2(x),$$

$$\textbf{Geometric Avg:} \quad p_{t,\beta}^{\text{geo}}(x) = \frac{1}{Z_t(\beta)} q_t^1(x)^{1-\beta} q_t^2(x)^{\beta}.$$

They derive weighted SDEs:

$$dx_t = (-f_t(x_t) + \sigma_t^2 \nabla \log p_t(x_t)) dt + \sigma_t dW_t,$$

and apply Sequential Monte Carlo (SMC) to correct sample trajectories. This framework generalizes classifier-free guidance (CFG) and improves inference-time controllability.

Thornton et al. (2025) introduce a novel framework that distills pretrained diffusion models into energy-parameterized models, where the score function is expressed as the gradient of a learned energy $s(x,t) = -\nabla_x E_\theta(x,t)$. To address training instability common in energy-based models, they propose a conservative projection loss that distills the score function from a pretrained teacher model:

$$\min_{\theta} \mathbb{E}_{p_t} \left[ \|\nabla E_\theta(x,t) + s_{\text{teacher}}(x,t)\|^2 \right].$$

This energy formulation enables the definition of new target distributions for sampling. Specifically, in compositional generation tasks, they define a composed distribution as the product of multiple submodels:

$$p(x) \propto \prod_i \exp(-E_i(x)) = \exp\left(-\sum_i E_i(x)\right),$$

which implies that the composed score is equivalent to the sum of individual scores:

$$\nabla \log p(x) = -\sum_i \nabla E_i(x) = \sum_i s_i(x).$$

This resembles the linear score composition used in prior works such as projective composition, but, unlike those methods, which plug the summed scores directly into the reverse diffusion equation.

Bradley et al. (2025) provide a theoretical foundation for *projective composition*. Assuming a *Factorized Conditional* structure, they show:

$$\nabla \log p(x) = \sum_i \nabla_x \log p_i(x) - (k-1)\nabla_x \log p_b(x).$$

where $p_b(x)$ is the background distribution (as in the unconditional model). This combination corresponds to a Bayesian fusion distribution:

$$p(x) \propto \frac{\prod_i p_i(x)}{p_b(x)^{k-1}}.$$

This enables direct reverse diffusion sampling without correction under appropriate assumptions. Furthermore, they generalize to feature space combinations, where orthogonal transformations $z = A(x)$ can preserve compositionality.

Skreta et al. (2025b) leverage the Itô density estimator to dynamically weight the score functions of different models, yielding a new generative vector field. Given $M$ diffusion models with score functions $\nabla \log q_t^i(x)$, SUPERDIFF defines:

$$u_\tau(x) = \sum_{i=1}^{M} \kappa_\tau^i(x) \nabla \log q_t^i(x).$$

In the OR mode (mixture of densities), the weights are defined by a softmax over the estimated log-densities:

$$\kappa_\tau^i(x) = \frac{\exp\left(T \cdot \log q_t^i(x) + \ell\right)}{\sum_j \exp\left(T \cdot \log q_t^j(x) + \ell\right)},$$

allowing the combined model to generate samples consistent with any of the component models. In the AND mode (intersection of densities), the weights are obtained by solving a linear system such that all log-densities evolve consistently:

$$\sum_{i=1}^{M} \kappa_\tau^i(x) \frac{d}{d\tau} \log q_t^i(x) \approx \text{constant across } i,$$

enforcing agreement among models and producing samples that satisfy all constraints simultaneously. Finally, the sampling process follows the reverse-time SDE:

$$dx_\tau = \left(-f_{1-\tau}(x_\tau) + g_{1-\tau}^2 u_\tau(x_\tau)\right)d\tau + g_{1-\tau}dW_\tau,$$

which yields samples representing either the union (OR) or the intersection (AND) of the underlying model distributions.

To further compare the differences in task adaptation, we also provide compositional methods with the same fine-grained input and conduct compositional experiments on both MIGC and InteractDiffusion. The results are compared with the AMDM method and presented in Table 6.

Table 6: Composition of MIGC and InteractDiffusion on the COCO-MIG Benchmark and HOI Detection Score

| Method | Instance Success Rate Avg ↑ | mIoU Score Avg ↑ | Default (Full) ↑ | Time (s) ↓ |
|---|---|---|---|---|
| Du et al. (2023) | 45.14 | 40.03 | 19.23 | 42.6 |
| Bradley et al. (2025) | 47.82 | 42.71 | 22.23 | 12.9 |
| Skreta et al. (2025a) | 46.35 | 41.25 | 21.74 | 13.6 |
| Skreta et al. (2025b) | 46.56 | 41.81 | 23.15 | 15.5 |
| InteractDiffusion(+MIGC) | **54.78** | **47.74** | 26.04 | **8.8** (single model 7.5s) |
| MIGC(+InteractDiffusion) | 49.78 | 43.69 | **31.40** | 9.4 |

From equation (4), we observe that the linear combination of different $\mathbf{z}_t$ from various models corresponds to the linear combination of their predicted noise, which is proportional to the score. Therefore, the linear combination in AMDM can be interpreted as a linear combination of scores, which is actually similar to the compositional methods. As shown in our aggregation ablation study of AMDM in Figure 5b, linear combination performs worse than spherical aggregation. This is because, in fine-grained generation, the data lies on a spherical manifold, and spherical aggregation better preserves generation quality. Although Du's method (Du et al., 2023) applies Langevin dynamics to improve sample quality after linear aggregation and achieves good results, it comes with substantial computational overhead. Similarly, SUPERDIFF (Skreta et al., 2025b) reduces manifold deviation from a methodological perspective and achieves favorable performance. However, in fine-grained generation scenarios, its "AND" mode requires solving a system of linear equations, which introduces additional overhead. Moreover, Bradley's work (Bradley et al., 2025) theoretically proves that linear score combination only yields high-quality results when the compositional conditions lie in orthogonal feature spaces. However, fine-grained generation typically involves nearly identical

conditions, which clearly violates this assumption, providing theoretical support for the observed experimental results.

Finally, we conduct a simple compositional generation experiment (2D Composition Mixture) to illustrate the limitations of AMDM in compositional generation. The experimental results are shown in Table 7. The reason for the failure of the AMDM algorithm is also quite simple: AMDM is only applicable under the approximate high-dimensional Gaussian distribution; for general distributions, spherical aggregation and deviation optimization fail. It is clear that AMDM differs significantly from other combination methods in terms of application scenarios.

Table 7: Comparison of AMDM and Du 2023 under a non-Gaussian distribution

| Method | ln(MMD) $\downarrow$ | LL $\uparrow$ | Var $\downarrow$ |
|---|---|---|---|
| AMDM | -3.51 | -2.43 | 0.032 |
| Du et al. (2023) | **-4.48** | **1.30** | **0.007** |

## F  EXPERIMENTAL DETAILS

### F.1  EVALUATION METRICS

**COCO-MIG Benchmark** (Zhou et al., 2024) assesses the enhancement in attribute metrics. The COCO-MIG Benchmark is based on the COCO-position layout, where each instance is assigned a color attribute, requiring the generated instances to satisfy both position and color constraints. The process includes sampling layouts from COCO and categorizing layouts into five levels (L2-L6) based on the number of instances. Then, colors are assigned to each instance, global prompts are constructed, and a test file containing 800 entries is generated. The COCO-MIG metrics primarily include Instance Success Rate and mIoU Score. The Instance Success Rate measures the probability of each instance being generated correctly, while mIoU Score calculates the average of the maximum IoU for all instances; if the color attribute is incorrect, the IoU value is set to 0. Since the MIG-Benchmark does not contain interactive prompts, we set the "action" input in the InteractDiffusion model to "and".

**FGAHOI** (Ma et al., 2023) can measure interaction controllability. We use the FGAHOI model based on the Swin-Tiny architecture, as an HOI detector to evaluate the model's ability to control interactions. HOI Detection Scores are categorized into two types: Default and Known Object. The Default setting is more challenging, as it requires distinguishing between irrelevant images.

**CLIP** (Hessel et al., 2021) is the CLIPScore of the generated images with captions of the image prompts.

### F.2  PARAMETER SETTINGS

**InteractDiffusion and MIGC.** The total sampling steps $T$ are set to 50, the aggregation step $s$ is set to 20, the weighting factor $w$ is set to 0.5 and the optimization steps $\eta_t^{\theta_1}$ and $\eta_t^{\theta_2}$ are both set to 0.3. We use InteractDiffusion v1.0, and MIGC is modified to use the DDIM sampling method to align with the same diffusion process.

**InteractDiffusion and IP-Adapter.** The total sampling steps $T$ are set to 10, the aggregation step $s$ is set to 5, the weighting factor $w$ is set to 0.5, ip scale is set to 0.8 and the optimization steps $\eta_t^{\theta_1}$ and $\eta_t^{\theta_3}$ are both set to 0.3. We utilized IP-Adapter based on SD1.5 while keeping InteractDiffusion unchanged.

**InteractDiffusion, MIGC and IP-Adapter.** The pretrained models for the three architectures remain consistent with those mentioned above. The total sampling step $T$ is set to 10, an aggregation step $s$ set to 5, and weight factors $w_1$ and $w_2$ set to 0.5 and 0.65. The optimization steps $\eta_t^{\theta_1}$, $\eta_t^{\theta_2}$, and $\eta_t^{\theta_3}$ are all simply set to 0.3, respectively.

### F.3 ADDITIONAL EXPERIMENTS

In order to further evaluate the style features, we additionally provide CLIP scores also on COCO validation set for reference. Table 8 shows a significant improvement in the aggregated style control from 0.533 to 0.581, nearly reaching the performance of IP-Adapter, also demonstrating the effectiveness of AMDM.

Table 8: CLIP scores for InteractDiffusion(+IP-Adapter).

| Method | CLIP $\uparrow$ |
|---|---|
| InteractDiffusion | 0.533 |
| InteractDiffusion(+IP-Adapter) | 0.581 |
| IP-Adapter | 0.588 |

In the main experiments, we have used different diffusion versions to validate the generality, which are also within the same diffusion model ecosystem: the IP-Adapter model is based on SD1.5, while InteractDiffusion and MIGC are based on SD1.4. Additionally, we conducted experiments using SDXL-based InteractDiffusion and IP-Adapter. SDXL and SD 1.4/1.5 use different VAE encoders, meaning that they are in the different diffusion model ecosystem. Building on this, we additionally conduct experiments on models from the SDXL ecosystem to further validate the generality of the AMDM algorithm. We also used CLIP to evaluate style features. As shown in Table 9, the style control significantly improves from 0.545 to 0.597, nearly reaching the performance of IP-Adapter. Because AMDM operates exclusively on the intermediate latent variables $\mathbf{z}_t$ of the diffusion process rather than on the parameterization of noise prediction, it is intrinsically decoupled from both $\epsilon$-prediction and $v$-prediction. Consequently, as long as the latent $\mathbf{z}_t$ at the same time step $t$ is accessible, the method can be seamlessly paired with different prediction parameterizations and samplers without modifying model weights or training objectives, demonstrating broad applicability and portability.

Table 9: CLIP scores for InteractDiffusion(+IP-Adapter) based on SDXL.

| Method | CLIP $\uparrow$ |
|---|---|
| InteractDiffusion | 0.545 |
| InteractDiffusion(+IP-Adapter) | 0.597 |
| IP-Adapter | 0.613 |

Finally, we conduct experiments using InteractDiffusion based on SD1.4 and MIGC based on EDM sampling. Compared with DDPM, EDM adopts a different noise schedule, and therefore the two models belong to different diffusion model ecosystems. We use this experiment to validate the analysis in Section 3.1, which states that aggregation occurs within the same diffusion model ecosystem. As shown in Table 10, when the two models are not derived from the same SDE, the AMDM algorithm fails: the generated images exhibit poor quality and deviate from the data manifold, since the two data domains do not lie on the same $\mathcal{M}_t$, making aggregation inapplicable. This is because, as analyzed in Section 3.1, aggregation operations can only occur between models within the same diffusion model ecosystem. This constraint arises from mathematical properties and also constitutes a fundamental principle that other compositional methods must adhere to. In contrast, works such as (Biggs et al., 2024; Oh et al., 2025; Wang et al., 2025) impose an additional restriction, requiring identical denoising networks. From this perspective, AMDM does not impose specific constraints on the design of the denoising network. Given that many recent works build upon the same pretrained models for secondary development, these requirements are relatively easy to satisfy in practice, which endows AMDM with broad applicability and extends its potential to a wider range of application scenarios.

## G THEORETICALLY EXTENDING TO RECTIFIED FLOW

The AMDM algorithm was originally designed to address fine-grained generation in the Stable Diffusion community, but it can likewise be extended to methods based on Rectified Flow and the

Table 10: Quantitative results in different diffusion model ecosystems on the COCO-MIG benchmark and CLIP Score in InteractDiffusion(+MIGC).

| Method | Instance Success Rate (%) ↑ | | | | | | mIoU Score (%) ↑ | | | | | | CLIP Score ↑ | |
| --- | --- | --- | --- | --- | --- | --- | --- | --- | --- | --- | --- | --- | --- | --- |
| | $L_2$ | $L_3$ | $L_4$ | $L_5$ | $L_6$ | Avg | $L_2$ | $L_3$ | $L_4$ | $L_5$ | $L_6$ | Avg | Global | Local |
| InteractDiffusion | 37.50 | 35.62 | 35.31 | 30.62 | 34.16 | 34.06 | 32.98 | 31.63 | 30.82 | 28.29 | 30.40 | 30.40 | 31.09 | 27.56 |
| InteractDiffusion(+MIGC) | 20.03 | 18.67 | 18.45 | 15.62 | 16.29 | 18.13 | 17.64 | 16.38 | 15.60 | 13.25 | 14.98 | 15.03 | 14.77 | 13.91 |
| MIGC | 67.70 | 59.61 | 58.09 | 56.16 | 56.88 | 58.43 | 59.39 | 52.73 | 51.45 | 49.52 | 49.89 | 51.48 | 33.01 | 28.95 |

**DiT family.** The key is to extend Proposition 3.1 and Proposition 3.2 to the Rectified Flow setting. Rectified Flow defines the sampling procedure:

$$\frac{\mathrm{d}\mathbf{z}_t}{\mathrm{d}t} = \boldsymbol{v}_\theta(\mathbf{z}, t, y) \tag{42}$$

where $v_\theta(\mathbf{z}, t, y)$ is the trained Rectified Flow model. Then, the following proposition holds:

**Proposition G.1.** *Assume that the aggregated variable at time step $t$ is $\mathbf{z}_t'$. For the sampling step from $t$ to $t-1$, two rectified flow models $p_{\theta_1}$ and $p_{\theta_2}$ sample $\mathbf{z}_{t-1}^{\theta_1}$ and $\mathbf{z}_{t-1}^{\theta_2}$ respectively from (42). Then the approximate upper bound for the absolute value of the difference of the norms is:*

$$\left| \|\mathbf{z}_{t-1}^{\theta_1}\| - \|\mathbf{z}_{t-1}^{\theta_2}\| \right| \leq \sqrt{1 - 2\alpha(1 - \cos\varphi)}\delta, \quad \cos\varphi \geq 1 - \frac{\delta^2}{\|\mathbf{z}_{t-1}^{\theta_1}\|^2 + \|\mathbf{z}_{t-1}^{\theta_2}\|^2}, \tag{43}$$

*where $\varphi$ is the angle between $\mathbf{z}_{t-1}^{\theta_1}$ and $\mathbf{z}_{t-1}^{\theta_2}$, $\alpha \in [0, 1/4]$, $\delta \leq L_{\theta_1} L_{\mathcal{T}} + L_t$ and $L_{\theta_1}$ is the Lipschitz constant.*

*Proof.* The proof is similar to Proposition 3.1. For $\left| \|\mathbf{z}_{t-1}^{\theta_1}\| - \|\mathbf{z}_{t-1}^{\theta_2}\| \right|$, we likewise have:

$$\begin{aligned}
\left| \|\mathbf{z}_{t-1}^{\theta_1}\| - \|\mathbf{z}_{t-1}^{\theta_2}\| \right| &= \frac{\left| \|\mathbf{z}_{t-1}^{\theta_1}\|^2 - \|\mathbf{z}_{t-1}^{\theta_2}\|^2 \right|}{\|\mathbf{z}_{t-1}^{\theta_1}\| + \|\mathbf{z}_{t-1}^{\theta_2}\|} \\
&= \frac{\left| \langle \mathbf{z}_{t-1}^{\theta_1} + \mathbf{z}_{t-1}^{\theta_2}, \ \mathbf{z}_{t-1}^{\theta_1} - \mathbf{z}_{t-1}^{\theta_2} \rangle \right|}{\|\mathbf{z}_{t-1}^{\theta_1}\| + \|\mathbf{z}_{t-1}^{\theta_2}\|} \\
&\leq \frac{\|\mathbf{z}_{t-1}^{\theta_1} + \mathbf{z}_{t-1}^{\theta_2}\|}{\|\mathbf{z}_{t-1}^{\theta_1}\| + \|\mathbf{z}_{t-1}^{\theta_2}\|} \|\mathbf{z}_{t-1}^{\theta_1} - \mathbf{z}_{t-1}^{\theta_2}\|
\end{aligned} \tag{44}$$

For the term on the left-hand side, we also have:

$$\begin{aligned}
\frac{\|\mathbf{z}_{t-1}^{\theta_1} + \mathbf{z}_{t-1}^{\theta_2}\|}{\|\mathbf{z}_{t-1}^{\theta_1}\| + \|\mathbf{z}_{t-1}^{\theta_2}\|} &= \sqrt{\left( \frac{\|\mathbf{z}_{t-1}^{\theta_1} + \mathbf{z}_{t-1}^{\theta_2}\|}{\|\mathbf{z}_{t-1}^{\theta_1}\| + \|\mathbf{z}_{t-1}^{\theta_2}\|} \right)^2} \\
&= \sqrt{\frac{(\|\mathbf{z}_{t-1}^{\theta_1}\| + \|\mathbf{z}_{t-1}^{\theta_2}\|)^2 - 2\|\mathbf{z}_{t-1}^{\theta_1}\|\|\mathbf{z}_{t-1}^{\theta_2}\|(1 - \cos\varphi)}{(\|\mathbf{z}_{t-1}^{\theta_1}\| + \|\mathbf{z}_{t-1}^{\theta_2}\|)^2}} \\
&= \sqrt{1 - \frac{2\|\mathbf{z}_{t-1}^{\theta_1}\|\|\mathbf{z}_{t-1}^{\theta_2}\|}{(\|\mathbf{z}_{t-1}^{\theta_1}\| + \|\mathbf{z}_{t-1}^{\theta_2}\|)^2}(1 - \cos\varphi)} \\
&= \sqrt{1 - 2\alpha(1 - \cos\varphi)},
\end{aligned} \tag{45}$$

where $\alpha = \dfrac{\|\mathbf{z}_{t-1}^{\theta_1}\|\|\mathbf{z}_{t-1}^{\theta_2}\|}{\left( \|\mathbf{z}_{t-1}^{\theta_1}\| + \|\mathbf{z}_{t-1}^{\theta_2}\| \right)^2}$, $\alpha \in [0, \frac{1}{4}]$, and moreover $1 - 2\alpha(1 - \cos\varphi) \leq 1$. The key lies in the estimation of $\delta$:

$$\begin{aligned}
\mathbf{z}_{t-1}^{\theta_1} &= \mathbf{z}_t' - \boldsymbol{v}_{\theta_1}(\mathbf{z}_t', t, y_1), \\
\mathbf{z}_{t-1}^{\theta_2} &= \mathbf{z}_t' - \boldsymbol{v}_{\theta_2}(\mathbf{z}_t', t, y_2).
\end{aligned} \tag{46}$$

Thus:

$$
\begin{aligned}
\delta &= \|\mathbf{z}_{t-1}^{\theta_1} - \mathbf{z}_{t-1}^{\theta_2}\| \\
&= \|\boldsymbol{v}_{\theta_1}(\mathbf{z}_t', t, y_1) - \boldsymbol{v}_{\theta_2}(\mathbf{z}_t', t, y_2)\| \\
&= \|\boldsymbol{v}_{\theta_1}(\mathbf{z}_t', t, y_1) - \boldsymbol{v}_{\theta_1}(\mathbf{z}_t', t, y_2) + \boldsymbol{v}_{\theta_1}(\mathbf{z}_t', t, y_2) - \boldsymbol{v}_{\theta_2}(\mathbf{z}_t', t, y_2)\| \\
&\leq \|\boldsymbol{v}_{\theta_1}(\mathbf{z}_t', t, y_1) - \boldsymbol{v}_{\theta_1}(\mathbf{z}_t', t, y_2)\| + \|\boldsymbol{v}_{\theta_1}(\mathbf{z}_t', t, y_2) - \boldsymbol{v}_{\theta_2}(\mathbf{z}_t', t, y_2)\|, \\
&\leq L_{\theta_1}\|y_1 - y_2\| + \|\boldsymbol{v}_{\theta_1}(\mathbf{z}_t', t, y_2) - \boldsymbol{v}_{\theta_2}(\mathbf{z}_t', t, y_2)\| \\
&\leq L_{\theta_1} L_{\mathcal{T}} + L_t.
\end{aligned}
\tag{47}
$$

Here we also assume *functional proximity* under the DiT architecture: $|\boldsymbol{v}_{\theta_1}(\mathbf{z}_t, t, y) - \boldsymbol{v}_{\theta_2}(\mathbf{z}_t, t, y)| \leq L_t$, which concludes the proof.

For Proposition 3.2 in the Rectified Flow setting, the extension is straightforward: simply set $\boldsymbol{\mu}_{\theta_1}(\mathbf{z}_t^{\theta_1}, t, y_1) = \boldsymbol{v}_{\theta_1}(\mathbf{z}_t^{\theta_1}, t, y_1)$, since this is fully consistent with DDIM ODE-style sampling.

Therefore, in theory, the AMDM algorithm is successfully transferred to the RF-based DiT family.

# H  VISUAL RESULTS OF TABLE 3

To better demonstrate that diffusion models initially focus on generating coarse-grained features such as position, attributes, and style, while later stages emphasize quality and consistency, we visualize Table 3. The results are shown in Figure 9.

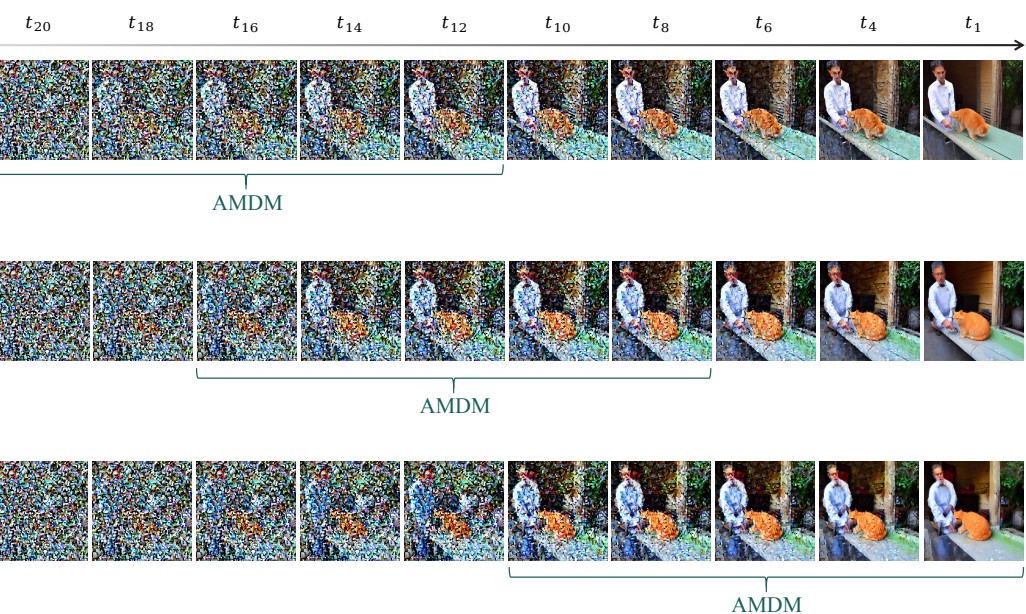

Figure 9: Visual Results of Table 3.

As we can see, diffusion models generally begin by focusing on generating position and shape, and only produce fine details in the final stages. For example, in the first row, the second sampling step already reveals the subject's shape and location. Next, when aggregation occurs in the latter half, image quality degrades and the picture becomes blurry. Notably, in the third row there is a clear jump between the 5th and 6th images, which severely affects the final result. We can anticipate that if aggregation happens only in the last few small steps, the generation quality will be even worse.

From Figure 10, it becomes even clearer that when aggregation occurs in the later stages, especially once the diffusion model begins focusing on detail generation, applying the aggregation algorithm at that point induces manifold deviation, causing a sharp deterioration in generation quality.

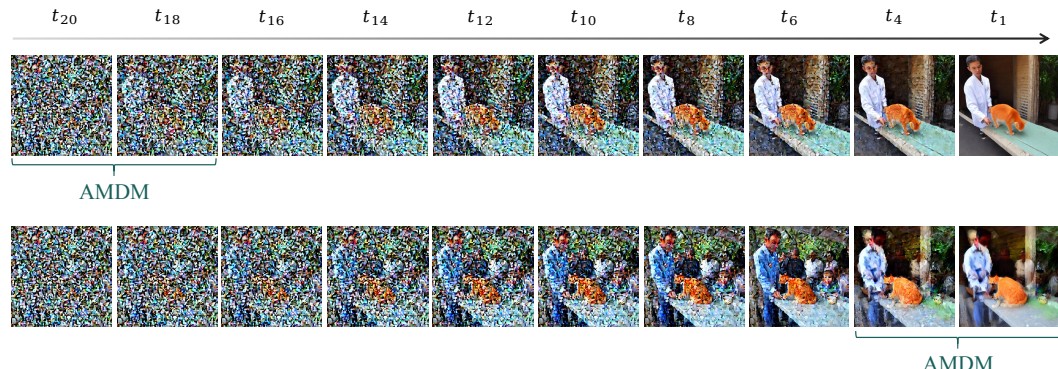

Figure 10: Different stages of the AMDM when aggregation steps are $s = 5$.

This further supports our conclusion that diffusion models initially focus on generating coarse-grained features such as position, attributes, and style, while later stages emphasize quality and consistency.

## I  LIMITATIONS AND FUTURE WORK

We propose a novel aggregation algorithm AMDM. This algorithm is specifically designed for fine-grained generation tasks within the same diffusion model ecosystem. The theoretical assumptions in this work (functional proximity, conditional proximity, and local Lipschitz continuity) are primarily tailored to fine-grained tasks, and this is empirically supported by our observations. These assumptions form the key distinction of AMDM from existing compositional diffusion approaches. Although it do not provide global guarantees, and we leave the development of more relaxed theoretical foundations to future work. In addition, the selection of the hyperparameter $\eta$ in the AMDM algorithm is largely heuristic. In future work, we will investigate dynamic adaptive selection strategies that are grounded in data-driven criteria or theoretical analysis to enhance the method's robustness and generalization.

## J  BROAD IMPACTS

We advocate against injecting harmful features into high-quality models to prevent the generation of socially harmful images.

## K  ADDITIONAL VISUAL RESULTS

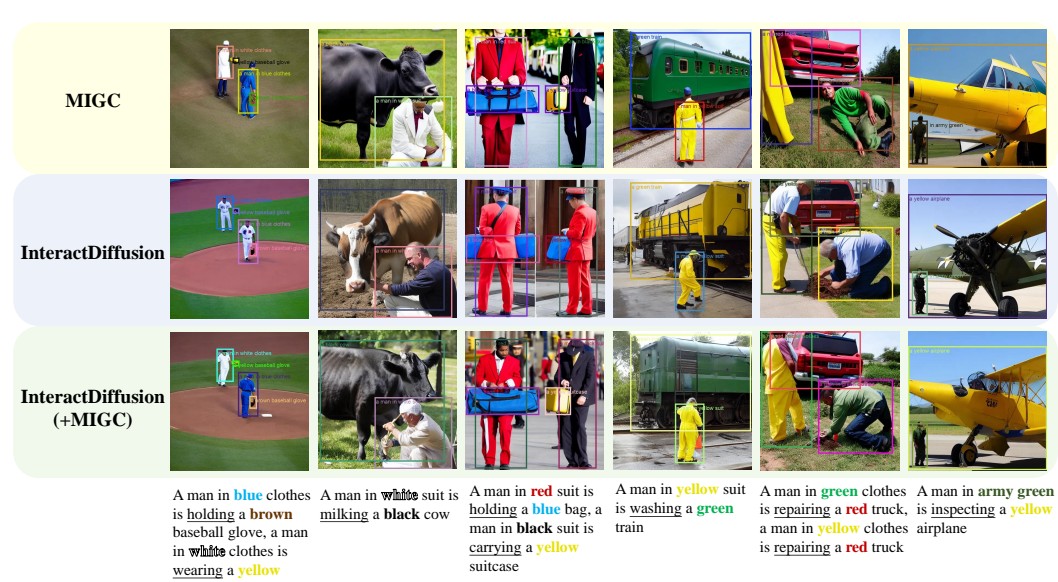

Figure 11: Additional visual results of aggregating MIGC into InteractDiffusion applying the AMDM algorithm.

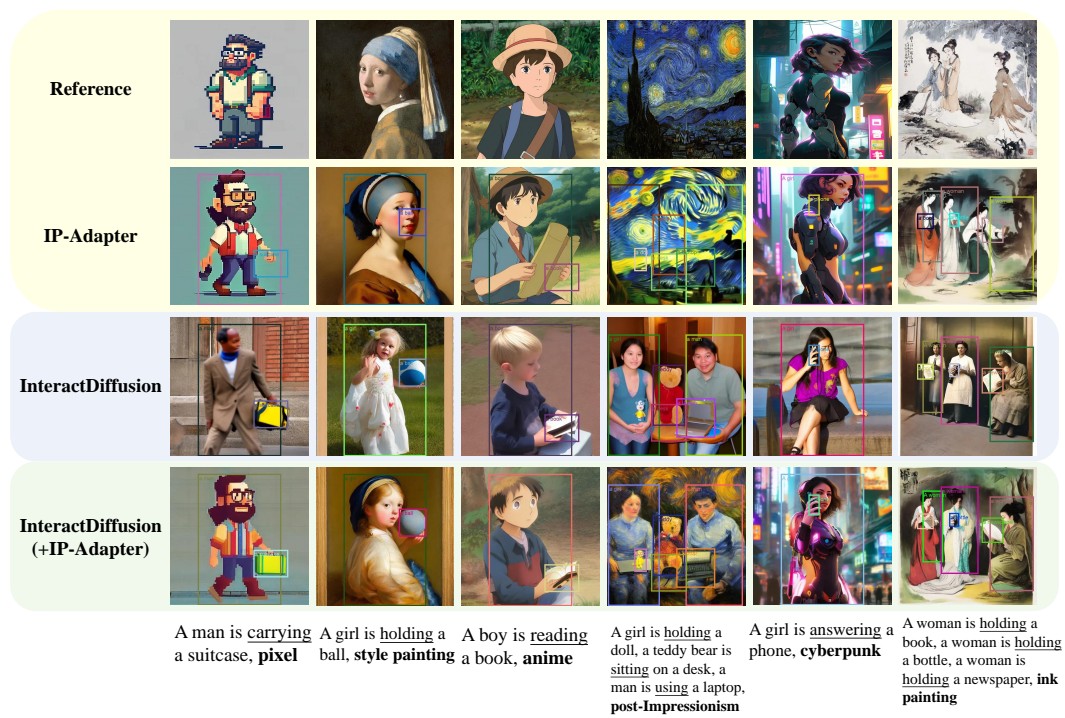

Figure 12: Additional visual results of aggregating IP-Adapter into InteractDiffusion applying AMDM algorithm.

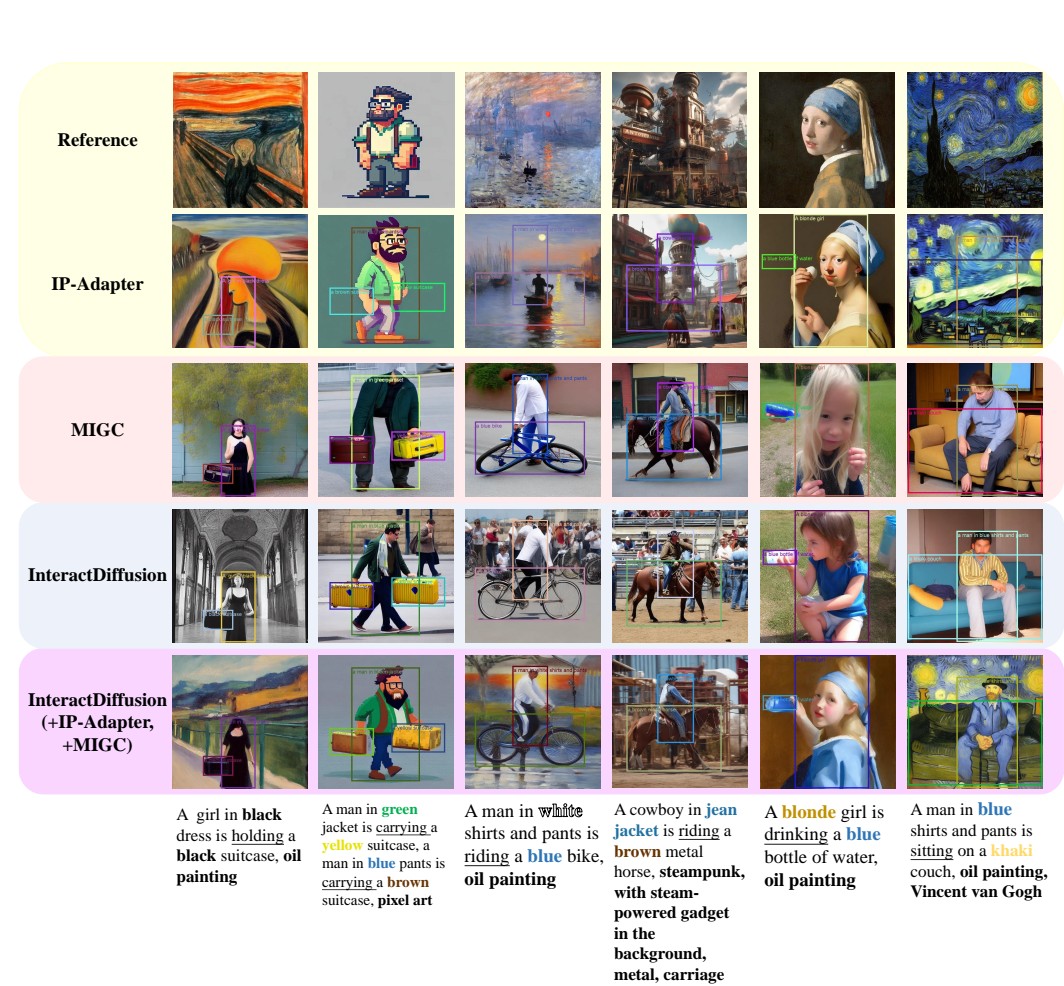

Figure 13: Additional visual results of aggregating MIGC and IP-Adapter into InteractDiffusion applying the AMDM algorithm.

## L    USE OF LLMS

We have carefully reviewed the paper for spelling, grammar, punctuation issues using LLMs.

