# OpenReview forum: "Improving Fine-Grained Control via Aggregation of Multiple Diffusion Models"
_ICLR.cc/2026/Conference — Submitted to ICLR 2026_

### Official Review · Reviewer_qUaq · 2025-10-27

**Soundness:** 2
**Presentation:** 3
**Contribution:** 2
**Rating:** 2
**Confidence:** 4

**Summary:**

This paper proposes a two-step process for combining multiple diffusion models. In the first step, the authors use Spherical Linear Interpolation (SLERP) to combine the models, and in the second step, they perform a descent procedure to move the combined output closer to the mean ( increasing probability of the sample ).

**Strengths:**

Propose novel method of Spherical Linear Interpolation for combining multiple diffusion models.

**Weaknesses:**

### Need Proofs for Claims / Additional Assumptions

**Lines 130–146:**
The assumption \(p_{\theta_1}, p_{\theta_2} \subset \mathcal{M}_0 \) implies that \( p_{\theta_1} \) and \( p_{\theta_2} \) are trained on the same data, or equivalently, that they are generative models of the same underlying data distribution.
However, this assumption holds **only if** both models are trained on data drawn from the same distribution. For example, if model A is trained on real images while model B is trained on paintings, the underlying data distributions differ, and hence the assumption \( p_{\theta_1}, p_{\theta_2} \subset \mathcal{M}_0 \) becomes invalid.

---

Can you provide a proof supporting the claim that additive or modified architectures only enhance control features?
This assertion appears **baseless** and **lacks supporting theoretical or empirical justification**. Consequently, the argument for a “same diffusion-model ecosystem” becomes **ill-defined and conceptually inconsistent**.

---

**Line 164:**
What do you mean by *“same task”*?

---

### Clarifications Needed

The motivation behind using *spherical interpolation* and *deviation optimization* is unclear.
It is also not evident how the parameters \( q_1, w_2, n_\theta \) are chosen.

There exist several approaches to aggregate diffusion models, many of which are derived from a probabilistic perspective under certain assumptions of independence (broadly referred to as *compositionality*).

### Comparison to Relevant Works

1. **Reduce, Reuse, Recycle:** [arXiv:2302.11552](https://arxiv.org/pdf/2302.11552)
2. **Compositionally:** [arXiv:2206.01714](https://arxiv.org/pdf/2206.01714)
3. **Conditional Independence Assumed:** [arXiv:2503.01145](https://arxiv.org/abs/2503.01145) — enforcing conditional independence
4. **Without Conditional Independence (Controllability):** [arXiv:2302.14368](https://arxiv.org/pdf/2302.14368)
5. [arXiv:2505.13213](https://arxiv.org/pdf/2505.13213)
6. **Algebraic Perspective:** [arXiv:2502.04549](https://arxiv.org/abs/2502.04549)

All these works rely on certain **underlying assumptions**, particularly regarding independence.
When such independence does not hold, an additional **weighting or hyperparameter term** is typically introduced to account for the violation.

---

### Example: Independence vs. Dependence

If conditional independence is assumed:

\[
p_{\theta}(x \mid y_1, y_2)
  = p_{\theta}(x \mid y_1)
  + p_{\theta}(x \mid y_2)
  - p_{\theta}(x \mid \varnothing)
\]

If independence **is not** assumed:

\[
p_{\theta}(x \mid y_1, y_2)
  = p_{\theta}(x \mid y_1)
  + w_1\, p_{\theta}(x \mid y_2)
  - w_1\, p_{\theta}(x \mid \varnothing),
\]

where \( w_1 \) controls the degree of independence between \( y_1 \) and \( y_2 \).

---

### Connecting to the Current Work

In line with the assumptions of the current work, suppose \( z_t \) is drawn from the distribution \( p(z_t) = p(z_t \mid \varnothing) \), and assume that \( z_t \subset \mathcal{M}_t \) for all \( t \in [0, T] \).
The objective can then be viewed as **linearly combining distributions** to maximize the probability of lying on the data manifold:

\[
\max_{a,b,c}\; p(a z^{(1)}_{t-1} + b z^{(2)}_{t-1} + c \mid \varnothing, z_t),
\]

such that the combination lies on the manifold of \( p(z_t) \).

Since the distribution is Gaussian, the maximum corresponds to its **mean**, reducing the sampling process to:

\[
a\, p_{\theta}(x \mid y_1, y_2)
  = p_{\theta}(x \mid y_1)
  + b\, p_{\theta}(x \mid y_2)
  + (1 - a - b)\, p_{\theta}(x \mid \varnothing).
\]

---

### Interpretation

Methods such as **ADAM** or **spherical interpolation** represent alternative interpolation strategies—implicitly introducing weighting factors rather than explicitly modeling independence.
In the current work, this combination is obtained via **spherical interpolation**, without direct access to \( p(z_t \mid \varnothing) \) (as in the classifier-free guidance formulation).
However, it remains unclear **why any \( s < T \)** should necessarily lie on a sphere.

---

### Broader Concern

For methods that do not assume any structural property of the data distribution, the introduction of a **hyperparameter** becomes unavoidable.
It is also unclear **how these hyperparameters should be selected**.

---

### Results and Evaluation

The reported results appear potentially misleading.
As the authors themselves claim, combining any two methods tends to improve performance over both individual methods.
Since the hyperparameters are tuned by the authors, the worst possible case corresponds to \( w_2 = 0 \).
Therefore, unless a **principled approach** for hyperparameter selection or a **comparison with existing aggregation methods** is provided, the results section offers **limited insight**.

**Questions:**

Addressing all the weaknesses will answer all my questions

**Details Of Ethics Concerns:**

### Need proofs for claims / additional assumptions

**Lines 130–146:**
The assumption $p_{\theta_1}, p_{\theta_2}$ $\in$  $M_0$ implies that $p_{\theta_1}$ and $p_{\theta_2}$ are trained on the same data, or equivalently, that they are generative models of the same underlying data distribution.  However, this assumption holds **only if** both models are trained on data drawn from the same distribution. For example, if model A is trained on real images while model B is trained on paintings, the underlying data distributions differ, and hence the assumption $p_{\theta_1}, p_{\theta_2} \in \mathcal{M}_0$ becomes invalid.

---

Can you provide a proof supporting the claim that additive or modified architectures only enhance control features?
This assertion appears **baseless** and **lacks supporting theoretical or empirical justification**. Consequently, the argument for a “same diffusion model” ecosystem becomes **ill-defined and conceptually inconsistent**.

---

**Line 164:**
What do you mean by *“same task”*?

---

### Clarifications needed

The motivation behind using *spherical interpolation* and *deviation optimization* is unclear.
It is also not evident how the parameters $q_1$, $w_2$, and $n_\theta$ are chosen.

There exist several approaches to aggregate diffusion models, many of which are derived from a probabilistic perspective under certain assumptions of independence (broadly referred to as *compositionality*).

---

### Comparison to relevant works

1. **Reduce, Reuse, Recycle:** [arXiv:2302.11552](https://arxiv.org/pdf/2302.11552)
2. **Compositionally:** [arXiv:2206.01714](https://arxiv.org/pdf/2206.01714)
3. **Conditional Independence Assumed:** [arXiv:2503.01145](https://arxiv.org/abs/2503.01145) — enforcing conditional independence
4. **Without Conditional Independence (Controllability):** [arXiv:2302.14368](https://arxiv.org/pdf/2302.14368)
5. [arXiv:2505.13213](https://arxiv.org/pdf/2505.13213)
6. **Algebraic Perspective:** [arXiv:2502.04549](https://arxiv.org/abs/2502.04549)

All these works rely on certain **underlying assumptions**, particularly regarding independence.
When such independence does not hold, an additional **weighting or hyperparameter term** is typically introduced to account for the violation.

If conditional independence is assumed:

$$p_{\theta}(x \mid y_1, y_2) = p_{\theta}(x \mid y_1)+ p_{\theta}(x \mid y_2)- p_{\theta}(x \mid \varnothing)$$

If independence is **not** assumed:

$$p_{\theta}(x \mid y_1, y_2)= p_{\theta}(x \mid y_1) + w_1\, p_{\theta}(x \mid y_2)- w_1\, p_{\theta}(x \mid \varnothing)$$

where $w_1$ controls the degree of independence between $y_1$ and $y_2$.

---

### Connecting to the current work

In line with the assumptions of the current work, suppose $z_t$ is drawn from the distribution $p(z_t) = p(z_t \mid \varnothing)$, and assume that $z_t \in \mathcal{M}_t$ for all $t \in [0, T]$.
The objective can then be viewed as **linearly combining distributions** to maximize the probability of lying on the data manifold:

$max_{a,b,c}$ $a\times p(z^{(1)}_{t-1}$
such that the combination lies on the manifold of $p(z_t)$.

Since the distribution is Gaussian, the maximum corresponds to its **mean**, reducing the sampling process to:

$$ p_{\theta}(x \mid y_1, y_2)= a \times p_{\theta}(x \mid y_1)+ b\times p_{\theta}(x \mid y_2)+ (1 - a - b)\times p_{\theta}(x \mid \varnothing)$$

Methods such as **ADAM** or **spherical interpolation** represent alternative interpolation strategies; implicitly introducing weighting factors rather than explicitly modeling independence.  In the current work, this combination is obtained via **spherical interpolation**, without direct access to $p(z_t \mid \varnothing)$ (as in the classifier-free guidance formulation).   However, it remains unclear **why any $s < T$** should necessarily lie on a sphere.

For methods that do not assume any structural property of the data distribution, the introduction of a **hyperparameter** becomes unavoidable.   It is also unclear **how these hyperparameters should be selected**.

### Results and evaluation

The reported results appear potentially misleading.  As the authors themselves claim, combining any two methods always improves the performance over both individual methods.  Since the hyperparameters are tuned by the authors, the worst possible case corresponds to $w_2 = 0$.  Therefore, unless a **principled approach** for hyperparameter selection or a **comparison with existing aggregation methods** is provided, the results section offers **limited insight**.

---

> ### Author Response · Authors · 2025-11-13
>
> We have carefully read the review comments and have tried our best to address the raised concerns:
>
> # 1. Need Proofs for Claims / Additional Assumptions
>
> 1. The assumption $p\_{\theta\_1}, p_{\theta\_2} \subset \mathcal{M}\_0$ does not hold.
>
> Here we believe there may be a misunderstanding. $p\_{\theta\_1}, p\_{\theta\_2}$ are probability distributions, while $\mathcal{M}\_0$ is a set, so the expression $p\_{\theta\_1}, p\_{\theta\_2} \subset \mathcal{M}\_0$ is not well-defined. Did you mean
> $D\_{0, y\_1}^{\theta\_1}, D\_{0, y\_2}^{\theta\_2} \subset \mathcal{M}\_0$?
>
> If it is the latter, then this is straightforward: $\mathcal{M}_0$ denotes the set of data encoded by the VAE, sampled from $p(z\_0)$. The distribution $p(z\_0)$ only depends on the VAE and is independent of $p\_{\theta\_1}, p\_{\theta\_2}$. In simple terms, $\mathcal{M}\_0$ is a large space that can cover very complex images, but due to the limited expressive power of $p\_{\theta\_1}, p\_{\theta\_2}$, they can only generate data sets such as $D\_{0, y\_1}^{\theta\_1}$ and $D\_{0, y\_2}^{\theta\_2}$, which are clearly subsets of $\mathcal{M}\_0$.
>
> In addition, this is not an assumption but a deduction. In our paper, the only assumptions we explicitly make are **condition proximity** and **function proximity**.
>
> 2. Proving that additive or modified architectures only enhance controllability. "Diffusion-model ecosystem" becomes ill-defined and conceptually inconsistent.
>
> In the paper, we define the diffusion model ecosystem as follows:
>
> 1. Theoretically, models must share the same diffusion paradigm (e.g., all based on DDPM);
> 2. Architecturally, they should fall into *Additive Architectures*, *Modified Architectures*, or *Preserved Architectures*.
>
> We are not fully sure what is meant by “proving that additive or modified architectures only enhance controllability.” Could the reviewer please clarify this point? Many text-to-image models in practice indeed achieve conditional control exactly by such means (e.g., fine-tuning, adding control modules, etc.). Thus, we do not fully understand what additional proof is being requested regarding “additive or modified architectures only enhancing controllability.”
>
> 3. What is meant by “same task”?
>
> By “the same task” we refer to the same fine-grained generation task: under fine-grained settings, different models share the same goal.
>
> # 2. Clarifications Needed
>
> 1. The motivation behind using spherical interpolation and deviation optimization is unclear.
>
> We propose Proposition 3.1, where we show that the norms of different data are almost equal and the angles between them are small. From this, we infer that they lie on a local spherical manifold. Deviation optimization is given by Proposition 3.2, which is introduced to project the data back to the manifold, and we provide a detailed proof of this.
>
> 2. Parameters chosen.
>
> We are not sure which parameters $q\_1$, $w\_2$ and $n\_\theta$ refer to. These symbols do not appear in our paper.
>
> # 3. Comparison to Relevant Works & Example: Independence vs. Dependence & Connecting to the Current Work
>
> We find it difficult to understand the precise question in this part and how the presented formulas and examples are intended to connect to our work. Could the reviewer kindly clarify the exact meaning or provide more context?
>
> Regarding the references you mentioned, we have analyzed and compared with them; the results are included in Table 6.
>
> # 4. Interpretation
>
> Propositions 3.1 and 3.2 provide detailed theoretical support for our motivation and design choices.
>
> # 5. Broader Concern
>
> We are not entirely sure what is meant by “methods that do not assume any structural properties of the data distribution.” Could the reviewer please clarify this concept or provide examples so that we can better address this concern?
>
> # 6. Results and Evaluation
>
> Again, we find it difficult to clearly understand the question here. Could the reviewer please clarify what is meant? For example, the statement that “the worst case actually corresponds to $w\_2 = 0$” does not appear in our paper.
>
> In addition, we have systematically compared AMDM with existing compositional methods in the appendix, and the experimental results show that AMDM achieves the best performance.
>
> # Conclusion
>
> We have carefully addressed the reviewer’s comments and sincerely hope that the reviewer can clarify the unclear points so that we can have a more in-depth discussion. If you have any further questions, please feel free to raise them and we will respond promptly.

---

> ### Comment · Reviewer_qUaq · 2025-11-16
> **Clarifications**
>
> 1. Incorrect claim $w_2 = 0$ does not appear in our paper.
>
> It does appear in your paper.  $w_1 \cdots w_{N}$ in the line 279 of the Algorithm 1. To clarify, here $w_2$ is the weight given for the guidance using second model, to provide further clarification it can be any weight $w_i$.
>
> Suppose, you want to show improvements in aggregation of two models, MIGC+ InteractDiffusion by setting $w_2=0$ as an hyperparameter it is equal to MIGC. So you would choose this hyper-paremeter to show at least improvement. given that you are free to choose hyper-parameter, as you offer no guidance on how to do it performamance of MIGC+ InteractDiffusion is always better than MIGC
>
> 2. Propositions 3.1 and 3.2 provide detailed theoretical support for our motivation and design choices.
>
> Propositions 3.1 and 3.2 do not provide the design choices mentioned in F.2.

---

> > ### Author Response · Authors · 2025-11-17
> > **Clarification reply 1/2**
> >
> > Thank you very much for your prompt reply. Let us continue addressing your questions:
> >
> > # 1. We thank the reviewer for the comments. I understand your concerns, but we believe this is a misunderstanding of AMDM. Let us clarify:
> >
> > **(1) "$w_2$ is guidance using second model."**
> >
> > $w_2$ is **not** the guidance using the second model! For aggregating two models, please refer to Eq. (6) in the main paper, where only $w$ is needed. The weights of the two models are controlled by $\frac{\sin((1-w)\varphi)}{\sin(\varphi)}$ and $\frac{\sin(w\varphi)}{\sin(\varphi)}$, rather than “$w_1$ and $w_2$.” The parameter $w_2$ appears only in the three-model aggregation case, controlling the guidance weight between the model aggregated in the first step (using $w_1$) and the third model. Therefore, in Algorithm 1 (line 273), we only choose $w_1, w_2, ..., w_{N-1}$.
> >
> > **(2) Suppose you want to show improvements in aggregation of two models, MIGC + InteractDiffusion. By setting “$w_2 = 0$” as a hyperparameter, it becomes equivalent to MIGC… given that you are free to choose hyperparameters…**
> >
> > Based on the explanation in (1), “freely choosing the hyperparameter” $w_2$ is **not valid**. Since aggregation between two models must follow spherical interpolation, their weights must satisfy the relationship $\frac{\sin((1-w)\varphi)}{\sin(\varphi)}$ and $\frac{\sin(w\varphi)}{\sin(\varphi)}$. Once one is fixed, the other is automatically determined. (For example, in linear interpolation, if one weight is 0.2, the other must be 0.8—otherwise it is not linear interpolation.)
> >
> > **(3) “Offer no guidance on how to do it; performance of MIGC + InteractDiffusion is always better than MIGC.”**
> >
> > This is incorrect. Aggregation does **not** always outperform the original model. A correct aggregation method—such as AMDM—is required. From the ablation study in Table 3, we can see that improper aggregation can lead to worse results; similarly, in Fig. 5(b), linear aggregation results in lower image quality than the original model.
> >
> > **(4) How to choose $w$?**
> >
> > The purpose of $w$ is to control the mixing ratio between models. Sometimes we want a model to exert stronger control over a particular attribute, in which case we increase that model’s $w$; this choice is **user-motivated**. In our experiments, $w$ was typically chosen around 0.5. We find this observation informative, and we expect this setting to work well in other scenarios.

---

> > > ### Author Response · Authors · 2025-11-17
> > > **Clarification reply 2/2**
> > >
> > > # 2. Propositions 3.1 and 3.2 provide detailed theoretical support for our motivation and design choices, not the design choices mentioned in F.2.
> > >
> > > Propositions 3.1 and 3.2 form the core motivation of AMDM. They **do not belong** to Appendix F.2 (experimental details) but appear in Appendix B, C, and D. Let us elaborate.
> > >
> > > **Proposition 3.1:**
> > >
> > > Unlike general compositional methods, AMDM is the **first** compositional generation method specifically designed for fine-grained tasks. Therefore, we first analyzed the special characteristics of fine-grained generation: all models share the same goal—generating samples under complex conditions—but due to capability limitations, each model can only control part of the attributes. Based on this, we infer that the conditional inputs of different models in fine-grained tasks are almost identical, and they tend to predict the same output.
> > >
> > > This leads to two assumptions: condition proximity and function proximity. Using these assumptions, we successfully derived Proposition 3.1, which states that the two variables have almost **identical magnitudes** and a **small angle** between them. Therefore, they lie on a **local spherical manifold**, which motivates the use of spherical aggregation. We also provide numerical analysis in Table 5 in the Appendix D.2 to verify the validity of the assumptions and Proposition 3.1. From the experimental results, AMDM significantly outperforms general compositional methods that were not designed for fine-grained tasks (Table 6).
> > >
> > > **Proposition 3.2:**
> > >
> > > The aggregated point is always slightly off the manifold, so we need to move it back toward the hypersphere to improve quality. In AMDM, this is done by moving the point along the gradient of the posterior distribution. Therefore, we propose Proposition 3.2 (deviation optimization); a detailed proof is provided in Appendix C, and ablation studies validate its effectiveness.
> > >
> > > The essence of the deviation optimization in AMDM lies in constraining the point to the hypersphere (data manifold). This idea aligns with the papers [1–3], where manifold correction methods (also leveraging certain gradient operator directions) are used to improve conditional control capability and sampling quality. These methods also provide theoretical support for the correction step proposed in this paper.
> > >
> > > The experiments also demonstrate the effectiveness of our optimization algorithm. As shown in the first and second rows of Fig. 5(b), when linear aggregation fails, incorporating deviation optimization **significantly improves** the generation quality and enables the model to produce meaningful images.
> > >
> > > **Overall**
> > >
> > > Our reasoning chain is: For this specialized fine-grained task, we analyze the nature of the problem → propose appropriate assumptions → derive Proposition 3.1, 3.2 and the AMDM algorithm → achieve very strong performance. From this design logic, AMDM is explicitly tailored for fine-grained generation in diffusion models; therefore, under this task, it significantly outperforms general compositional generation algorithms (Table 6). This is the primary advantage of AMDM.
> > >
> > > [1] Chung, et al. "Improving Diffusion Models for Inverse Problems using Manifold Constraints" NeurIPS 2022.
> > >
> > > [2] Chung, et al. "Diffusion Posterior Sampling for General Noisy Inverse Problems" ICLR 2023.
> > >
> > > [3] Yang, et al. "Guidance with Spherical Gaussian Constraint for Conditional Diffusion" ICML 2024.
> > >
> > > # Conclusion
> > >
> > > We hope that our responses can help you better understand AMDM. If you have any further questions, please feel free to raise them and we will respond promptly.

---

> ### Comment · Reviewer_qUaq · 2025-11-24
> **Final Comments**
>
> Thank you for providing explanation, I had missed the Appendix. E and Appendix D, which has answered my questions. I have raised my score to reflect it. Why not a higher score? Apart from details in F.2, the impact of hyper-parameters is not well studied, In essence, the question why do you need to set $w,\eta_t^{\theta_{1}}$, $, \eta_t^{\theta_{2}}$ to a certain value. and also as pointed by Reviewer MHdz The diffusion model ecosystem definition is unclear.

---

> > ### Author Response · Authors · 2025-11-24
> > **Final Comments Reply (1/2)**
> >
> > We sincerely appreciate your recognition of our AMDM method! We now proceed to address the specific concerns you raised:
> >
> > # 1. Selection of Parameters $w$, $\eta_1^{\theta_1}$, and $\eta_2^{\theta_2}$
> >
> > **(1) Selection of Parameter $w$:**
> >
> > We are grateful for the reviewer's suggestion. As previously mentioned, the parameter $w$ is **user-motivated**, where different choices of $w$ represent different control weights assigned to the models. Furthermore, following your suggestion, we conducted the following experiments using the classic InteractDiffusion(+MIGC) setup:
> >
> > Table 1. Experiments on InteractDiffusion(+MIGC) under different $w$ (Left two columns: Attribute Control; Right two columns: Action Control)
> > |$w$| ISR(avg) $\uparrow$ | mIoU Score(avg) $\uparrow$|Default(Full) $\uparrow$|Known Object(Full) $\uparrow$|
> > |-|-|-|-|-|
> > |0 (InteractDiffusion)|34.06|30.40|29.53|30.99|
> > |0.2|41.67|38.42|31.18|32.54|
> > |0.4|54.22|47.36|31.47|32.81|
> > |0.5|54.78|47.74|31.40|32.76|
> > |0.7|55.03|47.92|29.44|30.12|
> > |0.8|55.19|48.11|22.39|24.73|
> > |1.0 (MIGC)|55.46|48.53|16.87|17.84|
> >
> > Based on this experiment, we drew three main conclusions:
> >
> > 1.  $w$ controls the relative weights of the two models; a smaller $w$ yields results closer to InteractDiffusion, while a larger $w$ aligns more closely with MIGC.
> > 2.  When $w$ is near the midpoint (around 0.5), all performance metrics remain relatively stable.
> > 3.  When $w$ approaches the endpoints (0 or 1), the performance on the weaker aspect deteriorates rapidly. This is due to the properties of the sin function, where the actual fusion weights of the dominant model, $\frac{\sin((1-w)\varphi)}{\sin(\varphi)}$ and $\frac{\sin(w\varphi)}{\sin(\varphi)}$, drop sharply towards 0 as $w$ nears the extremes.
> >
> > Therefore, based on these findings, $w$ is a highly robust parameter. Within a certain range (e.g., around 0.5), users can adjust it freely to satisfy their desired generation results.
> >
> > **(2) Selection of Parameters $\eta_1^{\theta_1}$ and $\eta_2^{\theta_2}$:**
> >
> > This is an excellent question. In the original manuscript, the parameters $\eta_1^{\theta_1}$ and $\eta_2^{\theta_2}$ were determined based on the ablation study in Table 4; however, this selection was **not arbitrary**.
> >
> > The parameters $\eta_1^{\theta_1}$ and $\eta_2^{\theta_2}$ are derived from Proposition 3.2. One of the conceptual sources for this proposition is DSG [1] (as mentioned in the previous comment). Table 4 in the appendix of the DSG paper presents the step size parameter selection for various tasks, all of which are set to either 0.1 or 0.2, demonstrating high robustness. Inspired by this result, we hypothesized that under similar settings, our parameters $\eta_1^{\theta_1}$ and $\eta_2^{\theta_2}$ should also be around 0.2 (or on the order of $10^{-1}$). Consequently, we successfully pinpointed $\eta_1^{\theta_1},\eta_2^{\theta_2}=0.3$ through our ablation studies. Subsequent experiments further confirmed that these are also highly robust parameters; thus, we consistently set them to 0.3 and achieved excellent results.

---

> ### Author Response · Authors · 2025-11-24
> **Final Comments Reply (2/2)**
>
> # 2. Diffusion model ecosystem definition is unclear
>
> This is an excellent point, and we would like to provide a detailed explanation to clarify this concept.
>
> The "Diffusion model ecosystem" is a concept proposed in this paper, aiming to identify models that are compatible with the AMDM algorithm. The ecosystem imposes two primary constraints:
>
> 1.  **Theoretical Constraint:** The models must follow the same diffusion paradigm (e.g., DDPM);
> 2.  **Architectural Constraint:** They should satisfy the definitions of Additive Architectures, Modified Architectures, or Preserved Architectures.
>
> ***1. Why is the theoretical constraint necessary?***
>
> In fact, this constraint is trival. If the noise addition and denoising processes of two models differ, they obviously cannot be combined. In practice, all compositional methods implicitly assume this criterion by default, yet few provide detailed explanations. In the paragraph *"Do aggregation operations exist in different diffusion models?"* within *Section 3.1*, we **theoretically explain why** fine-grained compositional generation requires this constraint, thereby completing the theoretical foundation of compositional generation.
>
> ***2. Why is the architectural constraint necessary?***
>
> The purpose of the architectural constraint is to satisfy the unique hypothesis we proposed regarding fine-grained tasks: *functional proximity*. This requires models to exhibit similarity in their noise prediction behaviors.
>
> * **Additive Architectures:** Refer to adding plug-ins to the core SD model. Representative models include **ControlNet**, etc.
> * **Modified Architectures:** Refer to replacing or altering specific layers within the U-Net while retaining the overall structure. Examples include **DreamBooth** and **LoRA**.
> * **Preserved Architectures:** Refer to keeping the overall architecture intact while using new data to fine-tune weights. Representative methods include **Native Fine-tuning**.
>
> These three architectures cover almost all models currently in the diffusion community. Furthermore, we have verified through experiments in Appendix D that these architectures indeed exhibit similarity in noise prediction behavior.
>
> Additionally, some compositional methods [2-4] even require identical networks and weights. Compared to these approaches, AMDM is significantly more flexible.
>
> # Conclusion
>
> We thank the reviewer for their suggestion and will integrate the discussion above into the next version of the manuscript to further enhance it. We hope that our responses help you better understand AMDM. If you have any further questions, please feel free to raise them, and we will respond promptly.
>
> # References:
>
> [1] Yang, et al. "Guidance with Spherical Gaussian Constraint for Conditional Diffusion" ICML 2024.
>
> [2] Biggs, et al. "Diffusion Soup: Model Merging for Text-to-Image Diffusion Models", ECCV 2024.
>
> [3] Oh, et al. "DaWin: Training-free Dynamic Weight Interpolation for Robust Adaptation"， ICLR 2025.
>
> [4] Wang, et al. "Ensembling Diffusion Models via Adaptive Feature Aggregation", ICLR 2025.

---

### Official Review · Reviewer_WAAK · 2025-10-28

**Soundness:** 3
**Presentation:** 3
**Contribution:** 2
**Rating:** 4
**Confidence:** 3

**Summary:**

This paper tackles the problem of fine-grained conditional control in diffusion models. The proposed method, AMDM, aggregates multiple diffusion model scores via spherical interpolation, and employs a deviation optimization step to stabilize the aggregated score function. The framework enables users to combine the strengths of multiple fine-tuned diffusion models without requiring complex multi-capability datasets or multi-stage finetuning. Experiments on several conditional generation tasks demonstrate that AMDM can effectively integrate diverse capabilities from different models at inference time.

**Strengths:**

- The method is grounded in solid theoretical analysis, particularly regarding the confidence and reliability of aggregated diffusion scores.
- The empirical evaluation is thorough, and the comparisons clearly demonstrate how AMDM improves over baselines in multiple tasks.
- The motivation is practical and relevant: enabling reuse and integration of existing specialized diffusion models without retraining.

**Weaknesses:**

- Although the theoretical analysis is detailed, the paper may benefit from a simple controlled or toy example to help intuitively illustrate the effect of score aggregation and deviation optimization.
- The evaluation primarily focuses on three types of models (MIGC, InteractDiffusion, and IP-Adapter). While the results are encouraging, a broader range of conditional diffusion methods or application settings would better support the generality of the method.
- One of the key claims — that diffusion models initially prioritize feature generation before later refining image quality and consistency — is mainly supported by a single ablation (Table 3). Additional analysis or diagnostic visualization would help strengthen this conclusion.

**Questions:**

1. Regarding spherical interpolation and Equation (6):

    Could the authors clarify the geometric intuition or assumption behind applying spherical aggregation to score vectors? Is the assumption that these score fields locally reside on a shared spherical manifold, or is the spherical constraint introduced primarily for normalization and stability?

2. Computation overhead:

    What is the computational cost of AMDM compared to running a single diffusion model?

---

> ### Author Response · Authors · 2025-11-13
>
> Thank you very much for your valuable suggestions. We have carefully read your comments and will address your concerns one by one:
>
> ### 1. **Weakness 1**: Intuitive demonstration of the effects of spherical aggregation and deviation optimization.
>
> From the ablation study in Figure 5b, the effects of spherical aggregation and bias optimization can be clearly observed: comparing Rows 1 and 3, spherical aggregation produces meaningful images compared with linear aggregation, and the more aggregation steps used, the better the results. Comparing Rows 1 and 2, it is very evident that deviation optimization can transform low-quality images into high-quality ones.
>
> ### 2. **Weakness 2**: Generality of AMDM.
>
> In addition to experiments on three types of models (MIGC, InteractDiffusion, and IP-Adapter) under SD1.4/1.5, we also conducted experiments on SDXL, as shown in Table 9. Furthermore, AMDM can theoretically be extended to RF. In **Appendix G** of the original paper, we added the section *Theoretically Extending to Rectified Flow* (highlighted in red), where we generalize the theoretical foundations of AMDM to RF, along with proofs.
>
> However, the RF community currently has fewer studies on layout, attribute, interaction, and style compared to SD, and RF models often require significantly more computational resources and stronger hardware. We plan to conduct more extensive experiments with greater computational resources in the future to demonstrate the applicability of AMDM to RF and promote fine-grained generation within the RF community.
>
> ### 3. **Weakness 3**: Visualization analysis of ablation.
>
> Thank you very much for the suggestion. In **Appendix H** of the revised version, we added visualizations for Table 3 (highlighted in red), which make the correctness of Claim 3 more evident.
>
> ### 4. **Question 1**: Spherical interpolation and Equation (6).
>
> This is the core of our AMDM method. Let us explain the logic behind the spherical aggregation formula in detail.
>
> Unlike general compositional methods, AMDM is the **first** compositional generation method specifically designed for fine-grained tasks. Therefore, we first analyzed the special characteristics of fine-grained generation: all models share the same goal—generating samples under complex conditions—but due to capability limitations, each model can only control part of the attributes. Based on this, we infer that the conditional inputs of different models in fine-grained tasks are almost identical, and they tend to predict the same output.
>
> This leads to two assumptions: **condition proximity** and **function proximity**. Using these assumptions, we successfully derived Proposition 3.1, which states that the two variables have almost **identical magnitudes** and a **small angle** between them. Therefore, they lie on a **local spherical manifold**, which motivates the use of spherical aggregation.
>
> We also provide numerical analysis in Table 5 in the appendix to verify the validity of the assumptions and Proposition 3.1. From the experimental results, AMDM **significantly outperforms** general compositional methods that were not designed for fine-grained tasks (Table 6).
>
> ### 5. **Question 2**: Computational cost.
>
> We have provided this in Table 6 of the Appendix E in original paper.
>
> We sincerely appreciate the reviewer’s suggestions. Based on your questions, we encourage you to refer to Appendices E and F, where we provide a detailed analysis of the characteristics of the AMDM algorithm.
>
> ## Conclusion
>
> We truly hope that our responses help you better understand the AMDM method. If you have any further questions, please feel free to ask—we will respond immediately!

---

### Official Review · Reviewer_E8Pz · 2025-11-01

**Soundness:** 3
**Presentation:** 3
**Contribution:** 2
**Rating:** 6
**Confidence:** 3

**Summary:**

This paper introduces the Aggregation of Multiple Diffusion Models (AMDM) algorithm, a training-free method for improving fine-grained conditional control in image generation. AMDM works by aggregating features from multiple diffusion models within the same ecosystem to integrate their respective strengths. The key components are 1) Spherical aggregation 2)Deviation optimization. The authors demonstrate that AMDM significantly improves fine-grained control capabilities by integrating different models' strengths.

**Strengths:**

- The authors identify a genuine limitation in current models that they excel in specific aspects but struggle with others, and provide a solution without requiring retraining.
- The approach is backed by mathematical analysis of why aggregation works for models in the same diffusion ecosystem.
- Extensive experiments demonstrate clear improvements in both qualitative results and quantitative metrics.
- Unlike many compositional methods that introduce significant computational overhead, AMDM has minimal additional cost.

**Weaknesses:**

- While the authors provide theoretical justification, the assumptions (functional proximity, conditional proximity) are somewhat heuristic and don't offer global guarantees.
- AMDM only works for models within the same diffusion ecosystem, limiting its general applicability.
- While some comparisons with compositional methods are provided, a more comprehensive comparison with other training-free approaches would strengthen the paper.
- When aggregating models, there might be unintended interactions between different aspects that aren't fully explored.
- The optimization step size is selected empirically, and the authors acknowledge this as a limitation requiring future work.

**Questions:**

- How sensitive is AMDM to the choice of models within an ecosystem? Are there certain model combinations that work particularly well or poorly?
- (minor) What happens to the result if the positional information is not given? Would it still struggle to generate as specified from the text prompt or could this fine-grained control be coming from additional information, poisition.
- (minor) This is more like a question and just curiosity on my side. Could this fine-grained problem only exist within the open-source models? In other words, would the problem still exist in the models such as Sora?

---

> ### Author Response · Authors · 2025-11-13
>
> # Rebuttal 1/2
>
> Thank you very much for your valuable suggestions. We have carefully read your comments and will address your concerns one by one:
>
> ### 1. **Weakness 1**: Heuristic nature of the function proximity and condition proximity assumptions.
>
> Thank you for raising this excellent question! These two assumptions are indeed the core of the AMDM algorithm and fundamentally distinguish it from other compositional methods. Specifically, fine-grained generation requires solving the **same** generation task (although different models expressing different capabilities). Therefore, the conditional inputs must naturally be similar. If models within the same diffusion ecosystem have similar inputs, then their outputs are also similar. This eventually leads to our two central propositions 3.1 & 3.2: the sampling process lies on a **local spherical manifold**, which in turn justifies **spherical aggregation** and the **deviation optimization** algorithm.
>
> Overall, our reasoning chain is:
> For this specialized fine-grained task, we analyze the nature of the problem → propose appropriate assumptions → derive the AMDM algorithm → and achieve very strong performance. Moreover, in Table 5 of the Appendix, we numerically verify the reasonableness of these assumptions. From this design logic, AMDM is explicitly tailored for fine-grained generation in diffusion models; therefore, under this task, it significantly outperforms general compositional generation algorithms (Table 6)—this is the primary advantage of AMDM.
>
> ### 2. **Weakness 2**: Generality of AMDM.
>
> AMDM was originally designed to address fine-grained generation in the SD community, and within SD, AMDM is in fact *more general* than other compositional methods. First, all compositional methods also require consistency in the diffusion process. Second, many methods additionally require identical architectures (e.g., Diffusion Soup, Dawin), whereas AMDM only requires that models belong to the same model ecosystem, without further architectural constraints. Compared with other methods, AMDM not only provides a theoretical explanation for why fine-grained generation requires diffusion process consistency, but also offers a simpler and more efficient solution.
>
> In addition, AMDM can be extended to Rectified Flow! In Appendix G, we added a section titled *Theoretically Extending to Rectified Flow* (highlighted in red), where we generalize the theoretical foundations of AMDM to RF with accompanying proofs. Specifically, we again obtain that $\mathbf{z}^{\theta\_1}\_{t-1}$ and $\mathbf{z}^{\theta\_2}\_{t-1}$ have almost **equal magnitudes** and a very **small angle**, thus lie on a **local spherical manifold**, which makes spherical aggregation valid. Bias optimization also easily extends to the RF version.
>
> However, the RF community has relatively fewer studies on layout, attribute, interaction, and style compared to SD, and RF typically requires more resources and stronger hardware. We plan to conduct more extensive experiments with greater computational resources in the future to further demonstrate the applicability of AMDM to RF and help advance fine-grained generation within the RF community.
>
> ### 3. **Weakness 3**: Comparison with other training-free approaches would strengthen the paper.
>
> We greatly appreciate the suggestion. Due to space limitations in the main text, we placed most comparisons with compositional methods in Appendix E. In fact, AMDM is the **first** compositional algorithm specifically designed for fine-grained generation tasks. Therefore, it can naturally achieve significantly better results than general compositional methods—this is the core innovation of our paper. We believe more fine-grained compositional algorithms will emerge in the future, and our work provides a solid theoretical foundation for such future designs.
>
> ### 4. **Weakness 4**: Interaction effects not discussed.
>
> This is an astute observation. In line 361, we note that InteractDiffusion(+MIGC) *even surpasses* the original InteractDiffusion model in terms of interaction. We believe this is one characteristic of AMDM: under fine-grained compositional generation, different models can enhance each other’s expressiveness, and AMDM guarantees a lower bound of expression. This **collaborative nature** reflects the essence of fine-grained tasks, unlike many compositional generation settings where conditions may be **mutually exclusive**.

---

> ### Author Response · Authors · 2025-11-13
>
> # Rebuttal 2/2
>
> ### 5. **Weakness 5**: Selection of the step size parameter.
>
> This is an excellent question. In the original manuscript, the parameters $\eta_1^{\theta_1}$ and $\eta_2^{\theta_2}$ were determined based on the ablation study in Table 4; however, this selection was **not arbitrary**. The parameters $\eta_1^{\theta_1}$ and $\eta_2^{\theta_2}$ are derived from Proposition 3.2. One of the conceptual sources for this proposition is DSG [1]. Table 4 in the appendix of the DSG paper presents the step size parameter selection for various tasks, all of which are set to either 0.1 or 0.2, demonstrating high robustness. Inspired by this result, we hypothesized that under similar settings, our parameters $\eta_1^{\theta_1}$ and $\eta_2^{\theta_2}$ should also be around 0.2 (or on the order of $10^{-1}$). Consequently, we successfully pinpointed $\eta_1^{\theta_1},\eta_2^{\theta_2}=0.3$ through our ablation studies. Subsequent experiments further confirmed that these are also highly robust parameters; thus, we consistently set them to 0.3 and achieved excellent results.
>
> [1] Yang, et al. "Guidance with Spherical Gaussian Constraint for Conditional Diffusion" ICML 2024.
>
> ### 6. **Question 1**: Sensitivity to model choice.
>
> Through experiments with several distinct models that nevertheless belong to the same diffusion ecosystem, we found that different combinations can all produce strong results, whether A(+B) or B(+A). In the main text (line 290), we also emphasize that obtaining the best performance may require evaluating the inherent generative quality of the models themselves—but this is independent of AMDM.
>
> ### 7. **Question 2**: Whether AMDM partially relies on additional positional information.
>
> This is an excellent question. Precisely because of this concern, we intentionally used IP-Adapter for style guidance, since IP-Adapter does **not** include positional information. We wanted to observe whether abstract style features could still be effectively aggregated without positional cues. Clearly, AMDM achieved excellent results.
>
> ### 8. **Question 3**: Whether fine-grained control is only an issue in open-source models.
>
> The ultimate goal of generation is control. Open-source models usually have fewer parameters and smaller datasets, so they often excel only in specific types of control. The reviewer mentioned Sora, which is a very powerful closed-source model. Because of its large parameter count and vast training data, it naturally provides strong control across a broad set of attributes. However, regardless of model scale, such control bottlenecks always exist to some extent.
>
> Compositional generation methods—especially AMDM, which is tailored for fine-grained tasks—provide a form of **post-training enhancement**, allowing models developed by different institutions to complement each other and achieve more comprehensive control.
>
> # Conclusion
>
> We sincerely appreciate the reviewer’s suggestions. We truly hope the above responses help you better understand our AMDM method. If you have any further questions, please feel free to ask—we will respond immediately!

---

### Official Review · Reviewer_MHdz · 2025-11-01

**Soundness:** 3
**Presentation:** 3
**Contribution:** 2
**Rating:** 4
**Confidence:** 4

**Summary:**

The paper tackles the problem of fine-grained control in diffusion-based generative models that fail at nuanced, multi-conditional control ,e.g., spatial arrangement and style preservation. The authors propose a training-free algorithm called AMDM that aggregates latent representations from multiple diffusion models.  During inference, AMDM merge latent variables geometrically using spherical aggregation and a deviation optimization step. The authors conducted several empirical experiments to show improvements in attribute, style, and interaction controllability with AMDM.

**Strengths:**

- The perspective of leveraging existing diffusion models to address fine-grained, controllable generation in a training-free manner is an interesting direction.
- The paper is well-written and easy to follow; empirical performance and results demonstrate the promise of the proposed method.

**Weaknesses:**

- "Diffusion ecosystem" is mentioned across the paper as a prerequisite of AMDM but somewhat loosely defined, for example, how to quantitatively verify whether any two models belong to the same ecosystem is unclear.
- Evaluation scope is limited to certain derivatives of Stable Diffusion models. Throughout the experiments, the authors only picked a few classic SD1.4/1.5 and SDXL models, but did not examine whether the findings generalize to more recent model architectures based on DiT instead of U-Net, such as as SD3 or Flux.
- Comparing to other model composition methods in the literature, the proposed method only applies to high-dimensional Gaussian and fails to general distributions (as shown in Table 7)

**Questions:**

- Besides the comparisions between SD1.5 and SDXL, how sensitive is AMDM to broader model heterogeneity (schedulers or attention designs)? For example, you may conduct some experiments on models based on DiT to show the generality of your method?
- The weighting factor $w$ in Slerp is treated as hyperparameter, but lacks a principled selection method. Empirical experiments are conducted based on two/three model aggregation where $w$ is determined with ablations. Can you discuss more on how to systematically select these parameters, and especially with a set of models (beyond three typical models you selected), how your propose method work empirically?

---

> ### Author Response · Authors · 2025-11-13
> **Reply (1/2)**
>
> Thank you very much for your valuable suggestions. We have carefully read your comments and will address your concerns one by one:
>
> ### 1. **Weakness 1**: The diffusion model ecosystem is unclear.
>
> This is an excellent point, and we would like to provide a detailed explanation to clarify this concept.
>
> The "Diffusion model ecosystem" is a concept proposed in Section 3.1 *Analysis* of the paper, aiming to identify models that are compatible with the AMDM algorithm. The ecosystem imposes two primary constraints:
>
> 1. Theoretical Constraint: The models must follow the same diffusion paradigm (e.g., DDPM);
> 2. Architectural Constraint: They should satisfy the definitions of Additive Architectures, Modified Architectures, or Preserved Architectures.
>
> ***1. Why is the theoretical constraint necessary?***
>
> In fact, this constraint is trival. If the noise addition and denoising processes of two models differ, they obviously cannot be combined. In practice, all compositional methods implicitly assume this criterion by default, yet few provide detailed explanations. In the paragraph "Do aggregation operations exist in different diffusion models?" within Section 3.1, we theoretically explain why fine-grained compositional generation requires this constraint, thereby completing the theoretical foundation of compositional generation.
>
> ***2. Why is the architectural constraint necessary?***
>
> The purpose of the architectural constraint is to satisfy the unique hypothesis we proposed regarding fine-grained tasks: functional proximity. This requires models to exhibit similarity in their noise prediction behaviors.
>
> 1. Additive Architectures: Refer to adding plug-ins to the core SD model. Representative models include ControlNet, etc.
> 2. Modified Architectures: Refer to replacing or altering specific layers within the U-Net while retaining the overall structure. Examples include DreamBooth and LoRA.
> 3. Preserved Architectures: Refer to keeping the overall architecture intact while using new data to fine-tune weights. Representative methods include Native Fine-tuning.
>
> These three architectures cover almost all models currently in the diffusion community. Furthermore, we have verified through experiments in Appendix D that these architectures indeed exhibit similarity in noise prediction behavior. Additionally, some compositional methods [1-3] even require identical networks and weights. Compared to these approaches, AMDM is significantly more flexible.
>
> The reviewer asked how to quantitatively measure whether two diffusion models belong to the same ecosystem. We believe this is a very promising question, but also fundamentally difficult to solve from a technical perspective: it is extremely challenging to mathematically evaluate the correlation between two neural network architectures trained with different datasets and training parameters.
>
> [1] Biggs, et al. "Diffusion Soup: Model Merging for Text-to-Image Diffusion Models", ECCV 2024.
>
> [2] Oh, et al. "DaWin: Training-free Dynamic Weight Interpolation for Robust Adaptation"， ICLR 2025.
>
> [3] Wang, et al. "Ensembling Diffusion Models via Adaptive Feature Aggregation", ICLR 2025.
>
> ### 2. **Weakness 2 and Question 1**: Extending AMDM to RF.
>
> This is an excellent question! AMDM was originally designed to address fine-grained generation tasks within the current Stable Diffusion community, and it has already shown very strong performance. Of course, the AMDM method can be extended to Rectified Flow (RF) as well!
>
> In Appendix G of the revised version, we added a section titled *Theoretically Extending to Rectified Flow* (highlighted in red), where we extend the theoretical components of AMDM to RF, along with proofs. Specifically, we similarly obtain the conclusion that $\mathbf{z}^{\theta\_1}\_{t-1}$ and $\mathbf{z}^{\theta\_2}\_{t-1}$ have **almost equal magnitudes** and **a small angle** between them, thus lie on a **local spherical manifold**, making spherical aggregation valid. Moreover, the deviation optimization can also be easily extended to the RF version.
>
> However, research on layout, attribute, interaction, and style in the RF community is relatively limited compared to SD, and RF often involves higher resource consumption and more demanding hardware requirements. We plan to conduct more extensive experiments in the future, with more computational resources, to further demonstrate the applicability of AMDM to RF and promote the development of fine-grained generation in the RF community.

---

> ### Author Response · Authors · 2025-11-25
> **Reply (2/2)**
>
> ### 3. **Weakness 3**: Applicability of AMDM.
>
> As the reviewer pointed out, compared with other compositional generation methods, AMDM performs best mainly in high-dimensional Gaussian settings. Yet, we consider this to be a feature of the method. AMDM is specifically designed for fine-grained generation tasks in diffusion models, which is a very special and practically meaningful setting. Since we deliberately leverage the characteristics of fine-grained tasks, AMDM **significantly outperforms** general compositional generation methods, which do not take these characteristics into account. In Appendix Table 6, we also provide a detailed analysis of why AMDM can substantially surpass other compositional algorithms.
>
> ### 4. **Question 2**: Parameter selection in AMDM.
>
> We are grateful for the reviewer's suggestion. The parameter $w$ is **user-motivated**, where different choices of $w$ represent different control weights assigned to the models. Furthermore, following your suggestion, we conducted the following experiments using the classic InteractDiffusion(+MIGC) setup:
>
> Table 1. Experiments on InteractDiffusion(+MIGC) under different $w$ (Left two columns: Attribute Control; Right two columns: Action Control)
> |$w$| ISR(avg) $\uparrow$ | mIoU Score(avg) $\uparrow$|Default(Full) $\uparrow$|Known Object(Full) $\uparrow$|
> |-|-|-|-|-|
> |0 (InteractDiffusion)|34.06|30.40|29.53|30.99|
> |0.2|41.67|38.42|31.18|32.54|
> |0.4|54.22|47.36|31.47|32.81|
> |0.5|54.78|47.74|31.40|32.76|
> |0.7|55.03|47.92|29.44|30.12|
> |0.8|55.19|48.11|22.39|24.73|
> |1.0 (MIGC)|55.46|48.53|16.87|17.84|
>
> Based on this experiment, we drew three main conclusions:
>
> 1.  $w$ controls the relative weights of the two models; a smaller $w$ yields results closer to InteractDiffusion, while a larger $w$ aligns more closely with MIGC.
> 2.  When $w$ is near the midpoint (around 0.5), all performance metrics remain relatively stable.
> 3.  When $w$ approaches the endpoints (0 or 1), the performance on the weaker aspect deteriorates rapidly. This is due to the properties of the sin function, where the actual fusion weights of the dominant model, $\frac{\sin((1-w)\varphi)}{\sin(\varphi)}$ and $\frac{\sin(w\varphi)}{\sin(\varphi)}$, drop sharply towards 0 as $w$ nears the extremes.
>
> Therefore, based on these findings, $w$ is a **highly robust parameter**. Within a certain range (e.g., around 0.5), users can adjust it freely to satisfy their desired generation results.
>
> # Conclusion
> We thank the reviewer for their suggestion and will integrate the discussion above into the next version of the manuscript to further enhance it. We hope that our responses help you better understand AMDM. If you have any further questions, please feel free to raise them, and we will respond promptly.

---

### Author Response · Authors · 2025-12-02
**Summary for Area Chair**

Dear Area Chair,

We sincerely hope the following summary will help you quickly grasp the main contributions and our responses.

AMDM is the **first** proposed fine-grained compositional generation algorithm that utilizes different pre-trained diffusion models. It integrates the characteristics of various models, enabling fine-grained control over the generated images. The AMDM algorithm is concise, requiring only **a few mathematical operations**, demonstrating a significant advantage in **time efficiency and performance** for fine-grained generation tasks compared to other general compositional methods.

We sincerely thank the reviewers for their constructive feedback. We have organized our discussion into two stages in chronological order:

**1.Rebuttal Stage** (Response to the reviewers' initial reviews)：

1. Extending AMDM to Rectified Flow **(Reviewers MHdz, E8Pz and WAAK)**: AMDM was originally designed to address fine-grained generation tasks within the current Stable Diffusion community, and it has already shown very strong performance. We added **Appendix G** where we extend the theoretical components of AMDM to Rectified Flow, along with proofs.

2. The motivation of AMDM **(Reviewers E8Pz, WAAK and qUaq)**: We analyze the nature of the fine-grained generation tasks and propose appropriate assumptions **(Section 3.1)** → propose proposition 3.1 & 3.2 **(Section 3.2 and Appendix B & C & D)** → derive the AMDM algorithm → achieve very strong performance **(In addition to the main experiments, Appendices E & F are also included)**.

3. Comparison to Relevant Works **(Reviewer qUaq)**: We provide a detailed discussion of the literature mentioned by the reviewers in **Appendix E**.

4. Experiment Results **(Reviewer WAAK)**:

    (1) Computational cost: **Table 6 of Appendix E** in original paper.

    (2) Visualization analysis of ablation: We added **Appendix H**, which make the correctness of Claim more evident.

During this stage, Reviewers MHdz, E8Pz, and WAAK did not respond. Reviewer qUaq indicated that most issues had been resolved, **raised the score from 2 to 6 (Nov 24, 2025)**, and posed new questions.

**2.Discussion Stage** (Response to Reviewer qUaq's further questions, which also address the initial concerns of other reviewers):

1. Selection of Parameters $w$ **(Reviewers qUaq, MHdz)**: The parameter $w$ is **user-motivated**, where different choices of $w$ represent different control weights assigned to the models. We provide **Table 1** which indicates that $w$ is a highly robust parameter. Within a certain range (e.g., around 0.5), users can adjust it freely to satisfy their desired generation results.

2. Selection of Parameters $\eta_1^{\theta_1}$, and $\eta_2^{\theta_2}$ **(Reviewers qUaq, E8Pz)**: We addressed this issue in our response. These parameters were not chosen arbitrarily; instead, they were determined through empirical estimation and ablation studies **Table 4** in paper.

3. Diffusion model ecosystem definition is unclear **(Reviewers qUaq, MHdz)**: This concept appears in the third paragraph of the analysis in **Section 3.1** of the original paper. We also provided a detailed explanation of this concept and its motivation in our response.

In summary, we thank all reviewers for their constructive feedback, which has significantly strengthened the paper. We deeply regret that the incident with OpenReview led to the reversion of our improved scores from the rebuttal phase **(Reviewer qUaq 6 $\to$ 2)**. We also sincerely appreciate the additional burden this places on the the Area Chair. We remain fully committed to refining the manuscript based on your valuable feedback, so we believe the paper will make a meaningful contribution to the community.

We greatly appreciate the Area Chair’s time and effort in handling this submission once again.

Best regards,

The Authors

---

### Meta-Review · Area_Chair_Hukf · 2026-01-06

**Summary:**

This paper introduces AMDM, a training-free algorithm that aggregates features from multiple diffusion models to achieve fine-grained control in image generation. The paper received mixed initial reviews with scores ranging from 2 to 6, though one reviewer explicitly raised their score from 2 to 6 after rebuttal before the OpenReview score reset incident. Three reviewers did not respond to the rebuttal. Reviewers acknowledged the practical value of combining specialized models without retraining and appreciated the theoretical grounding through spherical aggregation and deviation optimization. The authors addressed key concerns by extending the method to Rectified Flow, clarifying the motivation chain from task analysis to algorithm design, and providing comprehensive parameter sensitivity analysis showing that the weight parameter w is robust around 0.5. However, significant weaknesses remain including heuristic assumptions about function and condition proximity, limited generalization testing beyond three model types, and unclear systematic hyperparameter selection. Multiple reviewers noted that the paper primarily evaluates on MIGC, InteractDiffusion, and IP-Adapter without demonstrating broader applicability. While the ablations validate individual components, the integration of the two proposed modules feels somewhat disconnected. Given the lack of reviewer engagement during discussion and the under-reviewed status with only one clearly supportive score, I recommend rejecting this submission.

**Reviewer Concerns:**

Addressed: Extension to Rectified Flow (Appendix G), clarification of motivation chain from task analysis to algorithm derivation, parameter sensitivity analysis showing w robust around 0.5, and comprehensive comparison with related works (Appendix E).

Outstanding: Heuristic assumptions about function and condition proximity lack strong theoretical foundation. Generalization limited to three model types without broader validation. Three reviewers (MHdz, E8Pz, WAAK) did not respond to rebuttal, making it impossible to confirm satisfaction.

**Reviewer Scores:**

MHdz (6): No response to rebuttal; score uncertain

E8Pz (4): No response to rebuttal; score uncertain

WAAK (4): No response to rebuttal; score uncertain

qUaq (2→6): Explicitly raised to 6 after rebuttal (note: OpenReview score reset may show 2 in interface, 6 is reported by the authors, which cannot be verified now)

---

### Decision · Program_Chairs · 2026-01-26

Reject